# Parameterized Approximation Schemes for Fair-Range Clustering

**Zhen Zhang**[1,2], **Xiaohong Chen**[1,2,*], **Limei Liu**[1,2], **Jie Chen**[1,2],
**Junyu Huang**[3], **Qilong Feng**[3,2*]

[1]National Key Laboratory Cultivation Base for Data Intelligence and Smart Society,
Hunan University of Technology and Business, Changsha 410205, China
[2]Xiangjiang Laboratory, Changsha 410205, China
[3]School of Computer Science and Engineering, Central South University,
Changsha 410083, China
`zz@hutb.edu.cn, csu_cxh@163.com, seagullm@163.com,`
`chemjay@hnu.edu.cn, junyuhuang@csu.edu.cn, csufeng@mail.csu.edu.cn`

## Abstract

Fair-range clustering extends classical clustering formulations by associating each data point with one or more demographic labels. It imposes lower and upper bound constraints on the number of facilities opened for each label, ensuring fair representation of all demographic groups by the selected facilities. In this paper we focus on the fair-range $k$-median and $k$-means problems in Euclidean spaces. We give $(1 + \varepsilon)$-approximation algorithms with fixed-parameter tractable running times for both problems, parameterized by the numbers of opened facilities and demographic labels. For Euclidean metrics, these are the first parameterized approximation schemes for the problems, improving upon the previously known $O(1)$-approximation ratios given by Thejaswi et al. (KDD 2022).

## 1 Introduction

Clustering seeks to partition a given set of clients into disjoint, cohesive clusters. Among the many formalizations of clustering, the $k$-*median* and $k$-*means* problems are perhaps the most prevalent ones, owing to the concise nature of their descriptions. Given a set of clients and facilities in a metric space along with a positive integer $k$, the $k$-median and $k$-means problems aim to open at most $k$ facilities and connect each client to the nearest opened facility, such that the sum of the client-connection costs is minimized. In the $k$-median problem, the connection cost of each client is its distance to the corresponding facility, while in the $k$-means problem, it is the squared distance. Despite their seemingly simple definitions, the $k$-median and $k$-means problems are computationally challenging, and the development of their approximation algorithms continues to be a vibrant area of research. The current best approximation guarantees are the ratios of 2.613 [Gowda et al., 2023] for the $k$-median problem and 9 [Ahmadian et al., 2020] for the $k$-means problem.

The $k$-median and $k$-means problems are designed to maximize the similarity between clients and their corresponding facilities, allowing the opened facilities to be considered representative points for the client set. This understanding underscores the important roles that the $k$-median and $k$-means problems play in data summarization [Moens et al., 1999, Girdhar and Dudek, 2012]. However, algorithms developed for these problems can often yield unfair summarization of socioeconomic data, as they prioritize minimizing the clustering costs over considering the distribution of demographic labels (e.g., gender, age, race) associated with the opened facilities [Kay et al., 2015]. Driven by this

---

*Corresponding Authors

38th Conference on Neural Information Processing Systems (NeurIPS 2024).

rationale, there has been considerable interest in *fair-range clustering*. Given a set of data points associated with demographic labels, fair-range clustering extends classical clustering formulations by imposing lower and upper bound constraints on the number of opened facilities associated with each label, thereby ensuring fairness across different demographic groups.

An instance $(\ell, k, \mathcal{C}, \mathcal{F}, \vec{\alpha}, \vec{\beta}, \rho, \tau)$ of the fair-range clustering problem is specified by positive integers $\ell$ and $k$, sets $\mathcal{C}$ of clients and $\mathcal{F}$ of facilities in a metric space, vectors $\vec{\alpha} = (\alpha_1, \cdots, \alpha_\ell)$ and $\vec{\beta} = (\beta_1, \cdots, \beta_\ell)$ of $\ell$ positive integers satisfying $\alpha_t \leq \beta_t$ for each $t \in \{1, \cdots, \ell\}$, and integer $\rho \geq 1$, where each $f \in \mathcal{F}$ is associated with a set $\tau(f) \subseteq \{1, \cdots, \ell\}$ of demographic labels. A feasible solution to the instance is specified by a subset $\mathcal{H} \subseteq \mathcal{F}$ of no more than $k$ facilities satisfying $|\{f \in \mathcal{H} : t \in \tau(f)\}| \in [\alpha_t, \beta_t]$ for each $t \in \{1, \cdots, \ell\}$, and the cost of the solution is $\sum_{c \in \mathcal{C}} \min_{f \in \mathcal{H}} \delta^\rho(c, f)$, where $\delta$ is the distance function. The goal of the fair-range clustering problem is to identify a feasible solution with minimum cost.

The fair-range clustering problem is equivalent to the *fair-range $k$-median* (FkMed) problem when $\rho = 1$, and to the *fair-range $k$-means* (FkMeans) problem when $\rho = 2$. Despite their significance in applications requiring fair representations, the FkMed and FkMeans problems pose significantly greater computational challenges than classical clustering problems. As demonstrated by Thejaswi et al. [2021], designing polynomial-time algorithms with provable approximation guarantees for the FkMed and FkMeans problems is unlikely, as determining the existence of feasible solutions to their instances is NP-hard. For a simplified scenario where each facility is associated with a single demographic label, Thejaswi et al. [2021] showed that the FkMed and FkMeans problems can be reduced to the well known *matroid clustering* problem, which admits constant-factor approximation algorithms [Krishnaswamy et al., 2011, Li, 2011, Swamy, 2014, Friggstad and Zhang, 2016, Krishnaswamy et al., 2018], albeit with an $O(k)^{\ell-1}$ multiplicative overhead in algorithmic running time. Hotegni et al. [2023] latter gave an improved reduction to the matroid clustering problem that eliminates the $O(k)^{\ell-1}$ overhead. They further demonstrated that solving the FkMed and FkMeans problems in this simplified scenario can be achieved even more efficiently than solving matroid clustering problems, based on smaller-size linear programs.

In practical scenarios concerning clustering problems, the number of opened facilities (i.e., $k$) is often considerably smaller than the input size. As such, assuming $k$ to be small and treating it as a fixed parameter is a commonly used way for simplifying these problems, as exemplified in [Cohen-Addad et al., 2019, Goyal and Jaiswal, 2023, Chen et al., 2024, Jaiswal et al., 2024]. Unfortunately, the FkMed and FkMeans problems have been demonstrated to remain challenging even with this simplification: When both $k$ and the number of demographic labels (i.e., $\ell$) are fixed parameters, Thejaswi et al. [2022] established the W[2]-hardness of the FkMed and FkMeans problems, suggesting that exactly solving them in fixed-parameter tractable time (denoted as $\text{FPT}(k, \ell)$ time, meaning $n^{O(1)} h(k, \ell)$ for an input size of $n$ and a positive function $h$) is unlikely; Cohen-Addad et al. [2019] showed that the best possible approximation ratios of $\text{FPT}(k, \ell)$-time algorithms, even for the case where $\ell = 1$, cannot be better than $1 + 2e^{-1}$ for the FkMed problem and $1 + 8e^{-1}$ for the FkMeans problem. This matches the $\text{FPT}(k, \ell)$-time $(1 + 2e^{-1} + \varepsilon)$-approximation algorithm for the FkMed problem and $(1 + 8e^{-1} + \varepsilon)$-approximation algorithm for the FkMeans problem given by Thejaswi et al. [2022] for a simpler case considering only the lower bound constraint. Notably, the method for enumerating feasible constraint patterns given by Thejaswi et al. [2022] demonstrates that their algorithms can be effortlessly extended to accommodate the case involving both lower and upper bounds.

The negative result presented by Cohen-Addad et al. [2019] suggests that we cannot hope to approximate the FkMeans problem with a ratio better than $1 + 8e^{-1}$ and the FkMed problem with a ratio better than $1 + 2e^{-1}$ in $\text{FPT}(k, \ell)$ time when considering general metric spaces. However, this result does not preclude the possibility of achieving better approximations for these problems in more structured settings, such as Euclidean spaces, since the analysis in Cohen-Addad et al. [2019] is limited to general metrics. In this paper, we take the first step toward exploring the properties of Euclidean metrics for the FkMed and FkMeans problems. Our approach yields $\text{FPT}(k, \ell)$-time approximation schemes, as stated in Theorem 1 in Section 3.4.

## 1.1 Other Related Work

Due to the prevalence of Euclidean data in real-world applications involving clustering, significant attention has been devoted to developing algorithms that leverage the properties of Euclidean spaces.

Exploring these properties often leads to improved approximation guarantees. One such example can be found in [Cohen-Addad et al., 2022], where a $(2.406 + \varepsilon)$-approximation algorithm for the $k$-median problem and a $(5.912 + \varepsilon)$-approximation algorithm for the $k$-means problem in Euclidean spaces are proposed, improving upon the state-of-the-art approximation ratios for these problems under general metrics [Gowda et al., 2023, Ahmadian et al., 2020]. Furthermore, it has been shown that the Euclidean $k$-median and $k$-means problems admit *approximation schemes*[2] if $k$ is a fixed parameter and the opened facilities can be located arbitrarily [Kumar et al., 2010, Jaiswal et al., 2014, Bhattacharya et al., 2018, Ding and Xu, 2020]. These algorithms identify a subset of each client-cluster defined by an optimal solution and approximate the corresponding opened facility by the centroid of this subset. However, similar ideas are not applicable to the FkMed and FkMeans problems, as they involve finite sets of facilities and hard constraints on the labels of the opened facilities. In these cases, the centroids of the considered subsets are not guaranteed to be feasible as opened facilities.

Constraints on the number of opened facilities associated with different labels were first introduced by Hajiaghayi et al. [2010, 2012], inspired by budget considerations for the deployment of servers in content distribution networks. From then on, related clustering problems have been widely explored. When we are provided with an upper bound constraint and each facility is associated with a single label, the problems represent special cases of the matroid clustering problems and directly motivate research into the latter [Krishnaswamy et al., 2011]. For the lower-bounded case, there are FPT$(k, \ell)$-time approximation algorithms for the $k$-median and $k$-means cost functions (given by Thejaswi et al. [2022], as previously mentioned in Section 1), and a multi-swap local-search heuristic yields an $O(\ell)$-approximation for the $k$-median cost function if each facility has a single label [Thejaswi et al., 2021, Zhang et al., 2024].

In addition to imposing constraints on the distribution of labels associated with opened facilities, fair clustering has been extensively studied under various other settings that introduce different types of constraints. For example, *group fairness* requires each cluster to provide a fair representation of different demographic groups [Chierichetti et al., 2017, Bera et al., 2019, Bandyapadhyay et al., 2021, Dai et al., 2022, Wu et al., 2024], *proportional fairness* ensures that no subset of clients, of a given size, can find a closed facility that provides a lower connection cost to each of its members [Chen et al., 2019, Micha and Shah, 2020], *individual fairness* requires that the distance from each client to the nearest opened facility does not exceed a client-specified threshold [Jung et al., 2020, Mahabadi and Vakilian, 2020, Negahbani and Chakrabarty, 2021, Vakilian and Yalçiner, 2022, Ahmadi et al., 2022, Bateni et al., 2024], and *social fairness* aims to minimize the maximum clustering cost among groups of clients [Abbasi et al., 2021, Ghadiri et al., 2021, Makarychev and Vakilian, 2021, Goyal and Jaiswal, 2023, Abbasi et al., 2024].

## 1.2 Preliminaries

From now on, we consider an instance $\mathcal{I} = (\ell, k, \mathcal{C}, \mathcal{F}, \vec{\alpha}, \vec{\beta}, \rho, \tau)$ of the fair-range clustering problem satisfying $\rho \in \{1, 2\}$, $|\mathcal{C} \cup \mathcal{F}| = n$, and $\mathcal{C} \cup \mathcal{F} \subset \mathbb{R}^d$, along with a constant $\epsilon \in (0, 0.5)$. Given an integer $i \geq 1$ and a set $\mathcal{D}$, define $[i] = \{1, \cdots, i\}$, and let $[\mathcal{D}]^i$ be the Cartesian product $\underbrace{\mathcal{D} \times \cdots \times \mathcal{D}}_{i}$.

Given a point $x$ and a set $\mathcal{P}$ of points in an Euclidean space, let $\delta(x, \mathcal{P}) = \min_{p \in \mathcal{P}} ||x - p||$ denote the distance from $x$ to its nearest point in $\mathcal{P}$, and let $\delta^i(x, \mathcal{P}) = \min_{p \in \mathcal{P}} ||x - p||^i$ for each $i \geq 1$.

The following algebraic fact will be utilized in the analysis of the running times of our algorithms.

**Lemma 1** *Given two real numbers $s$ and $t$ greater than 1, we have $\log^t s \leq \max\{s, t^{O(t)}\}$.*

The following lemma extends triangle inequality[3] to squared Euclidean metrics.

**Lemma 2** *Given three points $x$, $y$, and $z$ in an Euclidean space and a real number $\gamma \in (0, 1]$, we have $||x - z||^2 \leq (1 + \gamma^{-1})||x - y||^2 + (1 + \gamma)||y - z||^2$.*

We will also consider the weighted version of the fair-range clustering problem, which can be defined as follows.

---

[2]An approximation scheme is a $(1 + \varepsilon)$-approximation algorithm, where $\varepsilon$ is an arbitrary small constant.
[3]Given three points $x$, $y$, and $z$ in an Euclidean space, we have $||x - z|| \leq ||x - y|| + ||y - z||$.

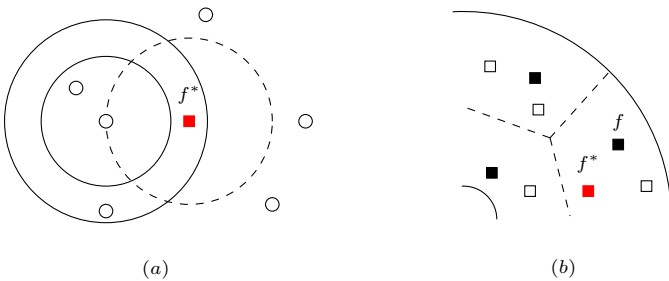

(a)                                                          (b)

Figure 1: $(a)$ The client nearest to the opened facility $f^*$ is taken as the leader, around which an annular search space is constructed; $(b)$ the center point $f$ is opened in our solution.

**Definition 1 (weighted fair-range clustering)** *An instance $(\ell, k, \mathcal{C}, \mathcal{F}, \vec{\alpha}, \vec{\beta}, \rho, \tau)$ of the fair-range clustering problem can be extended to its weighted version $(\ell, k, \mathcal{C}, \mathcal{F}, \vec{\alpha}, \vec{\beta}, \rho, \tau, w)$ by associating each client $c \in \mathcal{C}$ with a weight $w(c) \geq 1$. This extension modifies the cost of a feasible solution $\mathcal{H} \subseteq \mathcal{F}$ from $\sum_{c \in \mathcal{C}} \delta^\rho(c, \mathcal{H})$ to $\sum_{c \in \mathcal{C}} w(c) \delta^\rho(c, \mathcal{H})$.*

## 2 An Overview of Our Algorithms

The FPT$(k, \ell)$-time approximation algorithms for the FkMed and FkMeans problems given by Thejaswi et al. [2022] follow the framework outlined in [Cohen-Addad et al., 2019]. This framework identifies the nearest client to each facility opened in the considered optimal solution as a "leader" and introduces a set of annuli centered at each leader. Each annulus is defined such that its outer radius is $1 + \varepsilon$ times its inner radius. The framework then enumerates the annuli to identify those that contain the facilities corresponding to the leaders and selects the opened facilities within these annuli, as illustrated in Figure 1-$(a)$. Intuitively, the definition of the annuli and triangle inequality imply an upper bound on the distances from the selected facilities to the optimal ones. Building upon this insight, Thejaswi et al. [2022] utilized a submodular maximization method to select facilities to be opened within the annuli and demonstrated constant-factor approximation ratios.

We similarly base our algorithms on the framework proposed by Cohen-Addad et al. [2019]. Our approach focuses on exploring the properties of Euclidean metrics to further refine the selection range of opened facilities. Specifically, we partition each annulus into a set of smaller cells; for each facility opened in the optimal solution, we select the center point of the cell containing it as the facility to be opened, as shown in Figure 1-$(b)$. This process involves carefully balancing the number of cells, which affects the time required to identify the desired cells, against the sizes of the cells, which influence the distance from each facility opened in the optimal solution to the center point of the cell containing it, as well as our approximation ratio. We achieve this trade-off by constructing *nets* as defined below.

**Definition 2 ($\gamma$-net [Gupta et al., 2003])** *Given a density parameter $\gamma > 0$, a set $\mathcal{P} \subset \mathbb{R}^d$, and a subset $\mathcal{R} \subseteq \mathcal{P}$, we call $\mathcal{R}$ a $\gamma$-net of $\mathcal{P}$ if each $p \in \mathcal{R}$ satisfies $\delta(p, \mathcal{R} \setminus \{p\}) \geq \gamma$ and each $p \in \mathcal{P}$ satisfies $\delta(p, \mathcal{R}) \leq \gamma$.*

We partition the annular search space into cells using a set of nets for the facilities. The trade-off between the number and sizes of the cells can be managed by adjusting the density parameters (i.e., the parameter $\lambda$ in Definition 2). For each annulus centered around a leader and containing its corresponding facility (the one opened in the optimal solution), we estimate the demographic labels associated with this facility and identify the subset of facilities within the annulus that share these labels. A net is then constructed for this subset, using a density parameter carefully determined by the radius of the annulus. Given the Voronoi diagram defined by the net, Definition 2 suggests that the facility opened in the optimal solution is close to the center point of its corresponding Voronoi cell. By considering each member of all constructed nets as a candidate for an opened facility, we can ensure that the candidate set includes a subset closely approximating the optimal solution.

It remains to consider how to bound the running time within FPT$(k, \ell)$ time. The algorithms given by Cohen-Addad et al. [2019] and Thejaswi et al. [2022] start with constructing a coreset, that is, a small weighted subset of the client set whose distribution closely approximates that of the full set.

This facilitates the efficient identification of leaders by enumerating the coreset. In this paper, we face additional challenges in bounding the running time. For instance, when partitioning the annuli into cells, the number of cells can depend exponentially on the spatial dimension. To ensure that we can deal with the `FkMed` and `FkMeans` problems in high-dimensional Euclidean spaces within $\mathrm{FPT}(k, \ell)$ time, we map the considered instance to a space of $O(\log k + \log \log n)$ dimensions using a combination of the method for constructing coresets given by Chen [2009] and Johnson-Lindenstrauss transform. Combining this data-reduction technique with our net-based approach for selecting opened facilities, we give $\mathrm{FPT}(k, \ell)$-time $(1 + \varepsilon)$-approximation algorithms for the `FkMed` and `FkMeans` problems.

## 3 The Algorithms

We now present our algorithms for the `FkMed` and `FkMeans` problems. In Section 3.1, we introduce our data-reduction method for decreasing the size of the considered instance. In Section 3.2, we construct annular search spaces for the facilities to be opened, using the leaders from the client set. Section 3.3 details the construction of nets for the facilities, based on which we provide approximation schemes in low-dimensional spaces. Finally, in Section 3.4, we show how to combine the data-reduction method with the algorithms designed for low-dimensional spaces to deal with high-dimensional instances of the `FkMed` and `FkMeans` problems.

### 3.1 Data Reduction

In this section we map instance $\mathcal{I}$ to a smaller weighted instance in a low-dimensional space. As mentioned in Section 2, we achieve this using the coreset-construction method given by Chen [2009] and Johnson–Lindenstrauss transform, which are detailed in the following two lemmata.

**Lemma 3 (Chen [2009])** *Given a constant $\epsilon \in (0, 0.5)$, a set $\mathcal{P} \subset \mathbb{R}^d$, an integer $t > 0$, and an integer $\rho \in \{1, 2\}$, a subset $\mathcal{P}^\dagger \subseteq \mathcal{P}$ with a weight function $w : \mathcal{P}^\dagger \to [1, +\infty)$ satisfying $|\mathcal{P}^\dagger| \leq d(t\varepsilon^{-1} \log |\mathcal{P}|)^{O(1)}$ and $\sum_{p \in \mathcal{P}^\dagger} w(p) = |\mathcal{P}|$ can be constructed in $O(|\mathcal{P}|dt)$ time, such that each $\mathcal{H} \subset \mathbb{R}^d$ with $|\mathcal{H}| \leq t$ satisfies $\sum_{p \in \mathcal{P}^\dagger} w(p)\delta^\rho(p, \mathcal{H}) \in [1 - \epsilon, 1 + \epsilon] \sum_{p \in \mathcal{P}} \delta^\rho(p, \mathcal{H})$.*

**Lemma 4 (Johnson and Lindenstrauss [1984], Ailon and Chazelle [2009])** *Given a constant $\epsilon \in (0, 0.5)$ and a set $\mathcal{P} \subset \mathbb{R}^d$, we can construct a mapping $\varphi : \mathbb{R}^d \to \mathbb{R}^{\tilde{d}}$ satisfying $\tilde{d} = O(\epsilon^{-2} \log |\mathcal{P}|)$ and $||\varphi(p_1) - \varphi(p_2)|| \in [1, 1 + \epsilon]||p_1 - p_2||$ for each $p_1, p_2 \in \mathcal{P}$ in $O(d \log d) + (\epsilon^{-1} \log |\mathcal{P}|)^{O(1)}$ time.*

The following lemma is a stronger version of Johnson–Lindenstrauss transform, which preserves distances over a broader range through terminal embedding. Specifically, it modifies the condition "for each $p_1, p_2 \in \mathcal{P}$" in Lemma 4 to "for each $p_1 \in \mathcal{P}$ and $p_2 \in \mathbb{R}^d$".

**Lemma 5 (Narayanan and Nelson [2019])** *Given a constant $\epsilon \in (0, 0.5)$ and a set $\mathcal{P} \subset \mathbb{R}^d$, we can construct a mapping $\varphi : \mathbb{R}^d \to \mathbb{R}^{\tilde{d}}$ satisfying $\tilde{d} = O(\epsilon^{-2} \log |\mathcal{P}|)$ and $||\varphi(p_1) - \varphi(p_2)|| \in [1, 1 + \epsilon]||p_1 - p_2||$ for each $p_1 \in \mathcal{P}$ and $p_2 \in \mathbb{R}^d$ in $(|\mathcal{P}|d\epsilon^{-1})^{O(1)}$ time.*

It can be assumed that each mapping $\varphi : \mathbb{R}^d \to \mathbb{R}^{\tilde{d}}$ constructed by Lemma 4 and Lemma 5 is injective. Such an assumption is made without loss of generality: We can create duplicates of the points in $\mathbb{R}^{\tilde{d}}$ that have multiple preimages under $\varphi$. This ensures that we can always differentiate $\varphi(x)$ and $\varphi(y)$ for any two distinct points $x$ and $y$ in $\mathbb{R}^d$, even if $\varphi(x)$ and $\varphi(y)$ have identical values across all dimensions. Distinguishing the images of the points from $\mathbb{R}^d$ is essential in fair-range clustering problems because points with the same dimensional values can have different demographic labels.

Our data-reduction method, which combines Lemma 3, Lemma 4, and Lemma 5, is presented in Algorithm 1 and illustrated in Figure 2 (this figure outlines the processing flow for the clients). This algorithm first leverages Lemma 3 within the $O(\log n)$-dimensional space constructed by Lemma 4, such that the client set can be replaced with a coreset of size logarithmically dependent on $n$ and independent of $d$. Next, to reduce dimensions while preserving the distances between each client in the coreset and any facility, Algorithm 1 uses Lemma 5 with the coreset as input to construct an $O(\log k + \log \log n)$-dimensional space. The following lemma provides the performance guarantees of Algorithm 1.

---

**Algorithm 1:** The data-reduction method

---

**Input:** A constant $\epsilon \in (0, 0.5)$ and an instance $(\ell, k, \mathcal{C}, \mathcal{F}, \vec{\alpha}, \vec{\beta}, \rho, \tau)$ of the fair-range clustering problem satisfying $\mathcal{C} \cup \mathcal{F} \subset \mathbb{R}^d$

**Output:** A mapping $\varphi : \mathbb{R}^d \to \mathbb{R}^{\tilde{d}}$ and an instance $(\ell, k, \tilde{\mathcal{C}}, \tilde{\mathcal{F}}, \vec{\alpha}, \vec{\beta}, \rho, \tau, w)$ of the weighted fair-range clustering problem satisfying $\rho \in \{1, 2\}, \tilde{\mathcal{C}} \cup \tilde{\mathcal{F}} \subset \mathbb{R}^{\tilde{d}}$, $\tilde{\mathcal{F}} = \{\varphi(f) : f \in \mathcal{F}\}$, and $\tau(\varphi(f)) = \tau(f)$ for each $f \in \mathcal{F}$

**1** Let $\varphi_1 : \mathbb{R}^d \to \mathbb{R}^{d^\dagger}$ be the mapping constructed by Lemma 4 with $(\epsilon, \mathcal{C} \cup \mathcal{F})$ as the input;

**2** Let $\mathcal{C}^\dagger$ be the weighted set constructed by Lemma 3 with $(\epsilon, \{\varphi_1(c) : c \in \mathcal{C}\}, k, \rho)$ as the input, and let $w^\dagger : \mathcal{C}^\dagger \to [1, +\infty)$ be the corresponding weight function;

**3** Let $\varphi_2 : \mathbb{R}^{d^\dagger} \to \mathbb{R}^{\tilde{d}}$ be the mapping constructed by Lemma 5 with $(\epsilon, \mathcal{C}^\dagger)$ as the input;

**4** $\varphi \Leftarrow \varphi_2 \circ \varphi_1, \tilde{\mathcal{C}} \Leftarrow \{\varphi_2(c) : c \in \mathcal{C}^\dagger\}, \tilde{\mathcal{F}} \Leftarrow \{\varphi(f) : f \in \mathcal{F}\}$;

**5** **return** $\varphi : \mathbb{R}^d \to \mathbb{R}^{\tilde{d}}, (\ell, k, \tilde{\mathcal{C}}, \tilde{\mathcal{F}}, \vec{\alpha}, \vec{\beta}, \rho, \tau \circ \varphi^{-1}, w^\dagger \circ \varphi_2^{-1})$.

---

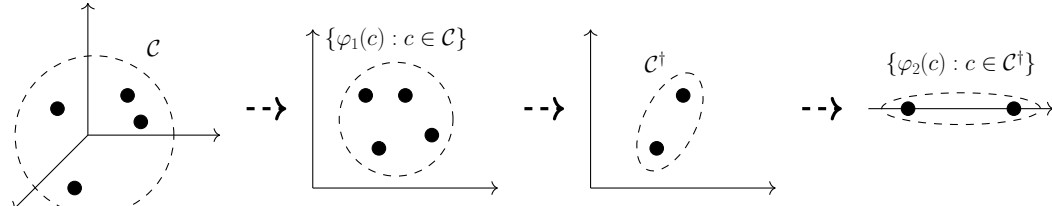

Figure 2: Given a set $\mathcal{C} \subset \mathbb{R}^d$ of clients, Algorithm 1 first maps it to $\mathbb{R}^{d^\dagger}$ using mapping $\varphi_1 : \mathbb{R}^d \to \mathbb{R}^{d^\dagger}$. It then constructs a coreset $\mathcal{C}^\dagger$ for $\{\varphi_1(c) : c \in \mathcal{C}\}$. Finally, the algorithm maps $\mathcal{C}^\dagger$ to $\mathbb{R}^{\tilde{d}}$ using mapping $\varphi_2 : \mathbb{R}^{d^\dagger} \to \mathbb{R}^{\tilde{d}}$.

**Lemma 6** *Given a constant $\epsilon \in (0, 0.5)$ and an instance $(\ell, k, \mathcal{C}, \mathcal{F}, \vec{\alpha}, \vec{\beta}, \rho, \tau)$ of the fair-range clustering problem with $\rho \in \{1, 2\}, |\mathcal{C} \cup \mathcal{F}| = n$, and $\mathcal{C} \cup \mathcal{F} \subset \mathbb{R}^d$, Algorithm 1 constructs a mapping $\varphi : \mathbb{R}^d \to \mathbb{R}^{\tilde{d}}$ and an instance $(\ell, k, \tilde{\mathcal{C}}, \tilde{\mathcal{F}}, \vec{\alpha}, \vec{\beta}, \rho, \tau, w)$ of the weighted fair-range clustering problem in $O(d \log d) + (nk\epsilon^{-1})^{O(1)}$ time, which satisfy the following properties:*

(i) $\sum_{c \in \tilde{\mathcal{C}}} w(c) = |\mathcal{C}|$,

(ii) $w(c) \geq 1$ *for each* $c \in \tilde{\mathcal{C}}$,

(iii) $|\tilde{\mathcal{C}}| \leq (k\epsilon^{-1} \log n)^{O(1)}$,

(iv) $\tilde{d} = \epsilon^{-O(1)}(\log k + \log \log n)$, *and*

(v) $\sum_{c \in \tilde{\mathcal{C}}} w(c)\delta^\rho(c, \{\varphi(f) : f \in \mathcal{H}\}) \in [1 - \epsilon, (1 + \epsilon)^{2\rho+1}] \sum_{c \in \mathcal{C}} \delta^\rho(c, \mathcal{H})$ *for each* $\mathcal{H} \subseteq \mathcal{F}$ *with* $|\mathcal{H}| \leq k$.

### 3.2 The Annular Search Spaces

In this section we construct $k$ annular search spaces, each corresponding to one of the $k$ facilities to be opened. We first introduce some notations. Let $\varphi : \mathbb{R}^d \to \mathbb{R}^{\tilde{d}}$ be the mapping and $\tilde{\mathcal{I}} = (\ell, k, \tilde{\mathcal{C}}, \tilde{\mathcal{F}}, \vec{\alpha}, \vec{\beta}, \rho, \tau, w)$ be the weighted instance constructed by Algorithm 1 with $(\epsilon, \mathcal{I})$ as the input, where $\tilde{\mathcal{C}} \cup \tilde{\mathcal{F}} \subset \mathbb{R}^{\tilde{d}}, \tilde{\mathcal{F}} = \{\varphi(f) : f \in \mathcal{F}\}$, and each $f \in \mathcal{F}$ satisfies $\tau(\varphi(f)) = \tau(f)$. Let $\tilde{\mathcal{H}}^* = \{f_1^*, \cdots, f_k^*\}$ be an optimal solution to $\tilde{\mathcal{I}}$, and let $opt = \sum_{c \in \tilde{\mathcal{C}}} w(c)\delta^\rho(c, \tilde{\mathcal{H}}^*)$ denote its cost. For each $i \in [k]$, let $\mathcal{L}_i = \{f \in \tilde{\mathcal{F}} : \tau(f) = \tau(f_i^*)\}$ denote the set of facilities that have the same set of demographic labels as $f_i^*$, and let $\tilde{\mathcal{C}}_i^* = \{c \in \tilde{\mathcal{C}} : \arg\min_{f \in \tilde{\mathcal{H}}^*} ||f - c|| = f_i^*\}$ be the cluster of clients defined by $f_i^*$. Given the lower bound constraint on the number of opened facilities, it may be the case that some facilities in $\tilde{\mathcal{H}}^*$ correspond to empty clusters. We thus define

**Algorithm 2:** The algorithm for constructing annular search spaces

**Input:** A constant $\epsilon \in (0, 0.5)$, an instance $\tilde{\mathcal{I}} = (\ell, k, \tilde{\mathcal{C}}, \tilde{\mathcal{F}}, \vec{\alpha}, \vec{\beta}, \rho, \tau, w)$ of the weighted fair-range clustering problem, and a positive integer $n$

**Output:** A collection $\mathbb{A}$

1  $\mathbb{A} \Leftarrow \emptyset$;
2  Let $\mathbb{D}$ be the power set of $[\ell]$;
3  **for** *each* $f \in \tilde{\mathcal{F}}$ **do**
4  $\quad$ Let $\eta(f)$ be an integer randomly and uniformly selected from $[k]$;

5  **for** *each* $(\mathcal{D}_1, \cdots, \mathcal{D}_k) \in [\mathbb{D} \backslash \{\emptyset\}]^k$, $k' \in [k]$, $(c'_1, \cdots, c'_{k'}) \in [\tilde{\mathcal{C}}]^{k'}$, *and*
$\quad$ $\delta' \in \{||c - f||^\rho : c \in \tilde{\mathcal{C}}, f \in \tilde{\mathcal{F}}\}$ **do**
6  $\quad$ **if** $|\{i \in [k] : t \in \mathcal{D}_i\}| \in [\alpha_t, \beta_t]$ *for each* $t \in [\ell]$ **then**
7  $\quad\quad$ **for** *each* $i \in [k]$ **do**
8  $\quad\quad\quad$ $\mathcal{L}'_i \Leftarrow \{f \in \tilde{\mathcal{F}} : \tau(f) = \mathcal{D}_i\}$;
9  $\quad\quad$ **for** *each* $i \in [k']$ *and* $j \in [\lceil \epsilon^{-2} \log n \rceil]$ **do**
10 $\quad\quad\quad$ $\mathcal{A}(i,j) \Leftarrow \{f \in \mathcal{L}'_i : ||f - c'_i||^\rho \in (\epsilon(1+\epsilon)^{j-1}\delta' n^{-1}, \epsilon(1+\epsilon)^j \delta' n^{-1}]\}$;
11 $\quad\quad\quad$ $\mathcal{A}(i,0) \Leftarrow \{f \in \mathcal{L}'_i : ||f - c'_i||^\rho \le \epsilon \delta' n^{-1}\}$;
12 $\quad\quad$ **for** *each* $(j_1, \cdots, j_{k'}) \in [[\lceil \epsilon^{-2} \log n \rceil] \cup \{0\}]^{k'}$ **do**
13 $\quad\quad\quad$ **for** *each* $i \in [k']$ **do**
14 $\quad\quad\quad\quad$ $\mathcal{A}_i \Leftarrow \{f \in \mathcal{A}(i, j_i) : \eta(f) = i\}$;
15 $\quad\quad\quad$ **for** *each* $i \in [k] \backslash [k']$ **do**
16 $\quad\quad\quad\quad$ **if** $\{f \in \mathcal{L}'_i : \eta(f) = i\} \ne \emptyset$ **then**
17 $\quad\quad\quad\quad\quad$ Let $\mathcal{A}_i$ be a singleton subset of $\{f \in \mathcal{L}'_i : \eta(f) = i\}$;
18 $\quad\quad\quad\quad$ **else**
19 $\quad\quad\quad\quad\quad$ $\mathcal{A}_i \Leftarrow \emptyset$;
20 $\quad\quad\quad$ $\mathbb{A} \Leftarrow \mathbb{A} \cup \{\{\mathcal{A}_1, \cdots, \mathcal{A}_k\}\}$;

21 **return** $\mathbb{A}$.

---

$\tilde{\mathcal{H}}_0^* = \{f \in \tilde{\mathcal{H}}^* : \tilde{\mathcal{C}}_i^* = \emptyset\}$ and $\tilde{\mathcal{H}}_1^* = \tilde{\mathcal{H}}^* \backslash \tilde{\mathcal{H}}_0^*$. Let $k^* = |\tilde{\mathcal{H}}_1^*|$. Without loss of generality, we can assume that $\tilde{\mathcal{H}}_1^* = \{f_1^*, \cdots, f_{k^*}^*\}$.

Following the framework given by Cohen-Addad et al. [2019], we select opened facilities from a set of annuli centered around a group of leaders from $\tilde{\mathcal{C}}$. For each $i \in [k^*]$, let $c_i$ denote the client from $\tilde{\mathcal{C}}$ nearest to $f_i^*$, that is, the leader corresponding to $f_i^*$. Let $\delta_{\max}^\rho = \max_{i \in [k^*]} \delta^\rho(c_i, f_i^*)$. For each $i \in [k^*]$ and $j \in [\lceil \epsilon^{-2} \log n \rceil]$, let $\mathcal{A}^*(i,j) = \{f \in \mathcal{L}_i : ||f - c_i||^\rho \in (\epsilon(1+\epsilon)^{j-1}\delta_{\max}^\rho n^{-1}, \epsilon(1 + \epsilon)^j \delta_{\max}^\rho n^{-1}]\}$ be the set of facilities from $\mathcal{L}_i$ located in an annulus centered around $c_i$, and let $\mathcal{A}^*(i,0) = \{f \in \mathcal{L}_i : ||f - c_i||^\rho \le \epsilon \delta_{\max}^\rho n^{-1}\}$. The definitions of $\mathcal{A}^*(i,j)$ and $\delta_{\max}^\rho$ imply the existence of an integer $j \in \{0, \cdots, \lceil \epsilon^{-2} \log n \rceil\}$ satisfying $f_i^* \in \mathcal{A}^*(i,j)$. Denote by $\tilde{\mathcal{A}}_i^*$ such a set $\mathcal{A}^*(i,j)$ containing $f_i^*$.

Our method for constructing annular search spaces is presented in Algorithm 2. Since the collection $\{\mathcal{A}^*(1,0), \cdots, \mathcal{A}^*(k^*, \lceil \epsilon^{-2} \log n \rceil)\}$ can be determined based on the values of $\{\mathcal{L}_1, \cdots, \mathcal{L}_k\}$, $k^*$, $\delta_{\max}^\rho$, and $\{c_1, \cdots, c_{k^*}\}$, Algorithm 2 enumerates all possible values of these parameters in step 5 to ensure that the collection can be captured. Given an integer $i \in [k^*]$ and the sets $\mathcal{A}^*(i,0), \cdots, \mathcal{A}^*(i, \lceil \epsilon^{-2} \log n \rceil)$, Algorithm 2 enumerates $[\lceil \epsilon^{-2} \log n \rceil] \cup \{0\}$ in step 12 to find the integer $j$ with $\tilde{\mathcal{A}}_i^* = \mathcal{A}^*(i,j)$. To avoid the case where the search spaces for the $k$ opened facilities intersect and the set of selected facilities contains duplicate elements, Algorithm 2 employs a color-coding technique to eliminate any potential intersections. Specifically, Algorithm 2 associates each facility $f \in \tilde{\mathcal{F}}$ with a random integer $\eta(f) \in [k]$ in step 4, and only selects facilities with $\eta(f) = i$ when constructing the $i$-th search space for each $i \in [k]$ in steps 14 and 17. The performance guarantees of this algorithm are presented in the following lemma.

**Lemma 7** *The collection $\mathbb{A}$ constructed by Algorithm 2 satisfies the following two properties:*

---

**Algorithm 3:** The algorithm for low-dimensional weighted instances

---

**Input:** A constant $\epsilon \in (0, 0.5)$, an instance $\tilde{\mathcal{I}} = (\ell, k, \tilde{\mathcal{C}}, \tilde{\mathcal{F}}, \vec{\alpha}, \vec{\beta}, \rho, \tau, w)$ of the weighted fair-range clustering problem, and a positive integer $n$

**Output:** A solution $\mathcal{H}^\dagger$ to $\tilde{\mathcal{I}}$

**1** $\mathbb{A} \Leftarrow \emptyset, \mathbb{H} \Leftarrow \emptyset$;

**2 for** *each $s \in [k^k]$* **do**

**3**     Let $\mathbb{A}'$ be the collection constructed by Algorithm 2 with $(\epsilon, \tilde{\mathcal{I}}, n)$ as the input;

**4**     $\mathbb{A} \Leftarrow \mathbb{A} \cup \mathbb{A}'$;

**5 for** *each $\{\mathcal{A}_1, \cdots, \mathcal{A}_k\} \in \mathbb{A}$ with $\mathcal{A}_i \neq \emptyset \, \forall i \in [k]$* **do**

**6**     **for** *each $i \in [k]$* **do**

**7**        **if** $|\mathcal{A}_i| = 1$ **then**

**8**           $\mathcal{S}_i \Leftarrow \mathcal{A}_i$;

**9**        **else**

**10**           Let $\mathcal{S}_i$ be the $\max_{x,y \in \mathcal{A}_i} \epsilon \|x - y\|$-net of $\mathcal{A}_i$ constructed by Lemma 8;

**11**     Let $\mathbb{H}'$ be the collection constructed by transforming each tuple in $\mathcal{S}_1 \times \mathcal{S}_2 \times \cdots \times \mathcal{S}_k$ into a set;

**12**     $\mathbb{H} \Leftarrow \mathbb{H} \cup \mathbb{H}'$;

**13 return** $\mathcal{H}^\dagger \Leftarrow \arg \min_{\mathcal{H} \in \mathbb{H}} \sum_{c \in \tilde{\mathcal{C}}} w(c) \delta^\rho(c, \mathcal{H})$.

---

(i) *With probability no less than $k^{-k}$, there exists a collection $\{\mathcal{A}_1, \cdots, \mathcal{A}_k\} \in \mathbb{A}$ satisfying $f_i^* \in \mathcal{A}_i \subseteq \mathcal{A}_i^*$ for each $i \in [k^*]$ and $\mathcal{A}_i \neq \emptyset$ for each $i \in [k] \backslash [k^*]$;*

(ii) *Given a collection $\{\mathcal{A}_1, \cdots, \mathcal{A}_k\} \in \mathbb{A}$, with $\mathcal{A}_i \neq \emptyset$ for each $i \in [k]$, and a tuple $(f_1, \cdots, f_k) \in \mathcal{A}_1 \times \mathcal{A}_2 \times \cdots \times \mathcal{A}_k$, we have $|\{i \in [k] : t \in \tau(f_i)\}| \in [\alpha_t, \beta_t]$ for each $t \in [\ell]$.*

## 3.3 The Algorithm in Low-Dimensional Spaces

As outlined in Section 2, solutions are constructed by extracting nets from the set of facilities. The following lemma presents a method for generating nets in low-dimensional Euclidean spaces.

**Lemma 8 (Har-Peled and Mendel [2006])** *Given a density parameter $\gamma > 0$ and a set $\mathcal{P} \subset \mathbb{R}^d$, a $\gamma$-net of $\mathcal{P}$ of size at most $\min\{|\mathcal{P}|, \gamma^{-d} \max_{p_1, p_2 \in \mathcal{P}} \|p_1 - p_2\|^d\}$ can be constructed in $|\mathcal{P}| \log |\mathcal{P}| 2^{O(d)}$ time.*

Our approach for solving the low-dimensional weighted instance $\tilde{\mathcal{I}}$ is built upon Algorithm 2 and Lemma 8, and is outlined in Algorithm 3. Since Algorithm 2 yields the desired search spaces with probability $k^{-k}$ (as given by the first property stated in Lemma 7), Algorithm 3 iteratively invokes it $k^k$ times, allowing the probability of successfully constructing the desired search spaces in at least one of the iterations to be boosted to a constant. Given $k$ sets $\mathcal{A}_1, \cdots, \mathcal{A}_k$ satisfying the first property stated in Lemma 7, Algorithm 3 constructs a net for each set of size greater than 1, and adds the members of the net to the candidate set of opened facilities. Finally, the algorithm constructs a set of feasible solutions to $\tilde{\mathcal{I}}$ based on the candidates for opened facilities, and returns the one with the minimum cost among them.

The following lemma says that Algorithm 3 yields a $(1 + O(\epsilon^{\frac{1}{\rho}}))$-approximation solution to $\tilde{\mathcal{I}}$ with high probability.

**Lemma 9** *The following event occurs with probability no less than $1 - e^{-1}$: Algorithm 3 yields a feasible $\mathcal{H}^\dagger$ to $\tilde{\mathcal{I}}$ satisfying $\sum_{c \in \tilde{\mathcal{C}}} w(c) \delta^\rho(c, \mathcal{H}^\dagger) < (1 + 4\epsilon) opt$ if $\rho = 1$ and $\sum_{c \in \tilde{\mathcal{C}}} w(c) \delta^\rho(c, \mathcal{H}^\dagger) < (1 + 9\sqrt{\epsilon}) opt$ if $\rho = 2$.*

By analyzing the time Algorithm 3 takes to construct the set of candidate solutions, as well as the size of this set, we can establish the following upper bound on the running time of Algorithm 3.

**Lemma 10** *Algorithm 3 runs in no more than $2^{(k\epsilon^{-1})^{O(1)} + k\ell} n^{O(1)}$ time.*

---

**Algorithm 4:** The algorithm in high-dimensional spaces

---

**Input:** A constant $\epsilon \in (0, 0.5)$ and an instance $\mathcal{I} = (\ell, k, \mathcal{C}, \mathcal{F}, \vec{\alpha}, \vec{\beta}, \rho, \tau)$ of the fair-range clustering problem satisfying $\mathcal{C} \cup \mathcal{F} \subset \mathbb{R}^d$
**Output:** A solution $\mathcal{H}^\ddagger$ to $\mathcal{I}$

1   Let $\varphi : \mathbb{R}^d \to \mathbb{R}^{\tilde{d}}$ be the mapping and $\tilde{\mathcal{I}} = (\ell, k, \tilde{\mathcal{C}}, \tilde{\mathcal{F}}, \vec{\alpha}, \vec{\beta}, \rho, \tau, w)$ be the weighted instance constructed by Algorithm 1 with $(\epsilon, \mathcal{I})$ as the input;
2   Let $\mathcal{H}^\dagger$ be the solution to $\tilde{\mathcal{I}}$ constructed by Algorithm 3 with $(\epsilon, \tilde{\mathcal{I}}, |\mathcal{C} \cup \mathcal{F}|)$ as the input;
3   **return** $\mathcal{H}^\ddagger \Leftarrow \{\varphi^{-1}(f) : f \in \mathcal{H}^\dagger\}$.

---

### 3.4 Extensions to High-Dimensional Spaces

We combine the data-reduction method given in Section 3.1 with the low-dimensional algorithm given in Section 3.3 to solve the FkMed and FkMeans problems in high-dimensional spaces, as detailed in Algorithm 4. Given a constant $\epsilon \in (0, 0.5)$ and an instance $\mathcal{I} = (\ell, k, \mathcal{C}, \mathcal{F}, \vec{\alpha}, \vec{\beta}, \rho, \tau)$, the algorithm starts with constructing a mapping $\varphi : \mathbb{R}^d \to \mathbb{R}^{\tilde{d}}$ and a small weighted instance $\tilde{\mathcal{I}} = (\ell, k, \tilde{\mathcal{C}}, \tilde{\mathcal{F}}, \vec{\alpha}, \vec{\beta}, \rho, \tau, w)$, where $\tilde{\mathcal{C}} \cup \tilde{\mathcal{F}} \subset \mathbb{R}^{\tilde{d}}$, $\tilde{\mathcal{F}} = \{\varphi(f) : f \in \mathcal{F}\}$, and $\tau(\varphi(f)) = \tau(f)$ for each $f \in \mathcal{F}$. It then constructs a solution to $\tilde{\mathcal{I}}$ and returns the set of preimages of the facilities opened in this solution under $\varphi$. The analysis of the performance guarantees of Algorithm 4 leads to the main result of this paper, as stated in Theorem 1.

**Theorem 1** *Given an instance $(\ell, k, \mathcal{C}, \mathcal{F}, \vec{\alpha}, \vec{\beta}, \rho, \tau)$ of fair-range clustering with $\mathcal{C} \cup \mathcal{F} \subset \mathbb{R}^d$ and $\rho \in \{1, 2\}$ along with a real number $\varepsilon \in (0, 1)$, there is a randomized $(1 + \varepsilon)$-approximation algorithm running in $O(d \log d) + 2^{(k\varepsilon^{-1})^{O(1)} + k\ell} n^{O(1)}$ time, where $n = |\mathcal{C} \cup \mathcal{F}|$.*

## 4 Conclusions

In this paper, we consider the FkMed and FkMeans problems for the case where the numbers of opened facilities and demographic labels are fixed parameters. Based on a combination of a data-reduction method and a space-partitioning approach for selecting opened facilities, we introduce $(1 + \varepsilon)$-approximation algorithms in Euclidean spaces, representing the first parameterized approximation schemes for the problems. Given that coreset-construction methods were known for many constrained clustering problems incorporating additional constraints on instances, such as those related to capacities [Cohen-Addad and Li, 2019], group fairness [Bandyapadhyay et al., 2021], and robustness [Huang et al., 2023], an interesting direction for future work is to extend our techniques to deal with the FkMed and FkMeans problems in similar constrained settings. Another promising avenue for exploration is to accelerate heuristic algorithms for the fair-range clustering problem, such as those given in [Thejaswi et al., 2022], using the data-reduction method proposed in this work.

## 5 Broader Impact

Our work deals with the fair-range clustering problem, providing algorithmic insights that can facilitate fair decision-making. While our algorithms have been shown to be "fair" according to specific definitions, it is essential to recognize that this fairness does not automatically warrant indiscriminate application. This underscores the need for careful consideration when implementing the algorithms proposed in this paper in real-world scenarios that prioritize fairness.

## Acknowledgements

This work was supported by Open Project of Xiangjiang Laboratory under Grant Nos. 22XJ03013 and 24XJJCYJ01003, National Natural Science Foundation of China under Grant Nos. 62202161, 62432016, 62172446, and 62202160, Natural Science Foundation of Hunan Province under Grant No. 2023JJ40240, and Central South University Research Programme of Advanced Interdisciplinary Studies under Grant No. 2023QYJC023.

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

## A Proof of Lemma 1

**Lemma 1** *Given two real numbers $s$ and $t$ greater than 1, we have $\log^t s \leq \max\{s, t^{O(t)}\}$.*

**Proof** If $t \geq \frac{\log s}{\log \log s}$, then we have $\log s \leq O(t \log t)$, and thus $\log^t s \leq t^{O(t)}$. If $t < \frac{\log s}{\log \log s}$, then we have $\log^t s < \log^{\frac{\log s}{\log \log s}} s = s$. Thus, Lemma 1 is true. $\square$

## B Proof of Lemma 2

**Lemma 2** *Given three points $x$, $y$, and $z$ in an Euclidean space and a real number $\gamma \in (0, 1]$, we have $||x - z||^2 \leq (1 + \gamma^{-1})||x - y||^2 + (1 + \gamma)||y - z||^2$.*

**Proof** Triangle inequality implies that $||x - z|| \leq ||x - y|| + ||y - z||$, and thus we have

$$||x - z||^2 \leq (||x - y|| + ||y - z||)^2$$

$$= ||x - y||^2 + ||y - z||^2 + 2\frac{1}{\sqrt{\gamma}}||x - y||\sqrt{\gamma}||y - z||$$

$$\leq ||x - y||^2 + ||y - z||^2 + \frac{1}{\gamma}||x - y||^2 + \gamma||y - z||^2.$$

This completes the proof of Lemma 2. $\square$

## C Proof of Lemma 6

**Lemma 6** *Given a constant $\epsilon \in (0, 0.5)$ and an instance $(\ell, k, \mathcal{C}, \mathcal{F}, \vec{\alpha}, \vec{\beta}, \rho, \tau)$ of the fair-range clustering problem with $\rho \in \{1, 2\}$, $|\mathcal{C} \cup \mathcal{F}| = n$, and $\mathcal{C} \cup \mathcal{F} \subset \mathbb{R}^d$, Algorithm 1 constructs a mapping $\varphi : \mathbb{R}^d \to \mathbb{R}^{\tilde{d}}$ and an instance $(\ell, k, \tilde{\mathcal{C}}, \tilde{\mathcal{F}}, \vec{\alpha}, \vec{\beta}, \rho, \tau, w)$ of the weighted fair-range clustering problem in $O(d \log d) + (nk\epsilon^{-1})^{O(1)}$ time, which satisfy the following properties:*

(i) $\sum_{c \in \tilde{\mathcal{C}}} w(c) = |\mathcal{C}|$,

(ii) $w(c) \geq 1$ *for each $c \in \tilde{\mathcal{C}}$,*

(iii) $|\tilde{\mathcal{C}}| \leq (k\epsilon^{-1} \log n)^{O(1)}$,

(iv) $\tilde{d} = \epsilon^{-O(1)}(\log k + \log \log n)$, *and*

(v) $\sum_{c \in \tilde{\mathcal{C}}} w(c)\delta^\rho(c, \{\varphi(f) : f \in \mathcal{H}\}) \in [1 - \epsilon, (1 + \epsilon)^{2\rho+1}] \sum_{c \in \mathcal{C}} \delta^\rho(c, \mathcal{H})$ *for each $\mathcal{H} \subseteq \mathcal{F}$ with $|\mathcal{H}| \leq k$.*

**Proof** Algorithm 1 constructs a mapping $\varphi_1 : \mathbb{R}^d \to \mathbb{R}^{d^\dagger}$ using Lemma 4 in step 1, a coreset $\mathcal{C}^\dagger$ along with the corresponding weight function $w^\dagger : \mathcal{C}^\dagger \to [1, +\infty)$ using Lemma 3 in step 2, and a mapping $\varphi_2 : \mathbb{R}^{d^\dagger} \to \mathbb{R}^{\tilde{d}}$ using Lemma 5 in step 3.

We begin by examining the first property of the output of Algorithm 1 stated in Lemma 6. We have

$$\sum_{c \in \tilde{\mathcal{C}}} w(c) = \sum_{c \in \mathcal{C}^\dagger} w^\dagger(c) = |\{\varphi_1(c) : c \in \mathcal{C}\}| = |\mathcal{C}|, \tag{1}$$

where the first step follows from the fact that $\tilde{\mathcal{C}} = \{\varphi_2(c) : c \in \mathcal{C}^\dagger\}$ (due to step 4 of Algorithm 1) and $w$ is the composite mapping $w^\dagger \circ \varphi_2^{-1}$ (due to step 5 of Algorithm 1), the second step follows from the fact that $\mathcal{C}^\dagger$ is the weighted set constructed by Lemma 3 with $\{\varphi_1(c) : c \in \mathcal{C}\}$ as the input set, and the last step is due to the assumption that the mappings constructed by Lemma 4 and Lemma 5 are injective. Equality (1) implies that the first property stated in Lemma 6 is true.

The second property stated in Lemma 6 follows directly from the fact that $w(c) = w^\dagger(\varphi_2^{-1}(c))$ for each $c \in \tilde{\mathcal{C}}$ (as established in step 5 of Algorithm 1) and $w^\dagger$ is a mapping to $[1, +\infty)$ (due to Lemma 3).

We now consider the third property stated in Lemma 6. This can be verified by

$$|\tilde{\mathcal{C}}| = |\mathcal{C}^\dagger| \leq d^\dagger(k\epsilon^{-1} \log |\mathcal{C}|)^{O(1)} = (k\epsilon^{-1} \log |\mathcal{C}|)^{O(1)} \log |\mathcal{C} \cup \mathcal{F}| = (k\epsilon^{-1} \log n)^{O(1)}, \tag{2}$$

where the first step is due to the fact that $\tilde{\mathcal{C}} = \{\varphi_2(c) : c \in \mathcal{C}^\dagger\}$ (as established in step 4 of the algorithm) and the assumption that each mapping constructed by Lemma 5 is injective, the second step follows from Lemma 3, and the third step is due to the fact that $\varphi_1 : \mathbb{R}^d \to \mathbb{R}^{d^\dagger}$ is the mapping constructed by Lemma 4 with $(\epsilon, \mathcal{C} \cup \mathcal{F})$ as the input.

Using inequality (2) and the fact that $\varphi_2 : \mathbb{R}^{d^\dagger} \to \mathbb{R}^{\tilde{d}}$ is the mapping constructed by Lemma 5 with $(\epsilon, \mathcal{C}^\dagger)$ as the input, we have

$$\tilde{d} = \epsilon^{-O(1)} \log |\mathcal{C}^\dagger| \le \epsilon^{-O(1)} \log(k\epsilon^{-1} \log n) = \epsilon^{-O(1)} (\log k + \log\log n),$$

which implies that the fourth property stated in Lemma 6 is true.

Given a subset $\mathcal{H} \subseteq \mathcal{F}$ with $|\mathcal{H}| \le k$, we have

$$\sum_{c \in \tilde{\mathcal{C}}} w(c)\delta^\rho(c, \{\varphi(f) : f \in \mathcal{H}\}) \in [1, (1+\epsilon)^\rho] \sum_{c \in \mathcal{C}^\dagger} w(c)\delta^\rho(c, \{\varphi_1(f) : f \in \mathcal{H}\})$$

$$\subseteq [1 - \epsilon, (1+\epsilon)^{\rho+1}] \sum_{c \in \mathcal{C}} \delta^\rho(\varphi_1(c), \{\varphi_1(f) : f \in \mathcal{H}\})$$

$$\subseteq [1 - \epsilon, (1+\epsilon)^{2\rho+1}] \sum_{c \in \mathcal{C}} \delta^\rho(c, \mathcal{H}),$$

where the first step follows from Lemma 5 and the fact that $\tilde{\mathcal{C}} = \{\varphi_2(c) : c \in \mathcal{C}^\dagger\}$ and $\varphi(f) = \varphi_2(\varphi_1(f))$ for each $f \in \mathcal{F}$ (due to step 4 of Algorithm 1), the second step follows from the fact that $\mathcal{C}^\dagger$ is a coreset of $\{\varphi_1(c) : c \in \mathcal{C}\}$ constructed by Lemma 3, and the last step is due to Lemma 4. This completes the proof of the last property stated in Lemma 6.

It remains to show the running time of Algorithm 1. Recall that the algorithm invokes Lemma 4 with $(\epsilon, \mathcal{C} \cup \mathcal{F})$, invokes Lemma 3 with $(\epsilon, \{\varphi_1(c) : c \in \mathcal{C}\}, k, \rho)$, and invokes Lemma 5 with $(\epsilon, \mathcal{C}^\dagger)$, where $\mathcal{C}^\dagger \subseteq \{\varphi_1(c) : c \in \mathcal{C}\} \subset \mathbb{R}^{d^\dagger}$. Combining this with inequality (2), we can express the running time of Algorithm 1 as

$$O(d \log d + |\mathcal{C}|d^\dagger k) + (\epsilon^{-1} \log |\mathcal{C} \cup \mathcal{F}|)^{O(1)} + (|\mathcal{C}^\dagger|d^\dagger \epsilon^{-1})^{O(1)} \le O(d \log d) + (nk\epsilon^{-1})^{O(1)},$$

as desired. $\qquad\square$

## D  Proof of Lemma 7

**Lemma 7** *The collection $\mathbb{A}$ constructed by Algorithm 2 satisfies the following two properties:*

(i) *With probability no less than $k^{-k}$, there exists a collection $\{\mathcal{A}_1, \cdots, \mathcal{A}_k\} \in \mathbb{A}$ satisfying $f_i^* \in \mathcal{A}_i \subseteq \mathcal{A}_i^*$ for each $i \in [k^*]$ and $\mathcal{A}_i \ne \emptyset$ for each $i \in [k] \backslash [k^*]$;*

(ii) *Given a collection $\{\mathcal{A}_1, \cdots, \mathcal{A}_k\} \in \mathbb{A}$, with $\mathcal{A}_i \ne \emptyset$ for each $i \in [k]$, and a tuple $(f_1, \cdots, f_k) \in \mathcal{A}_1 \times \mathcal{A}_2 \times \cdots \times \mathcal{A}_k$, we have $|\{i \in [k] : t \in \tau(f_i)\}| \in [\alpha_t, \beta_t]$ for each $t \in [\ell]$.*

**Proof** Algorithm 2 enumerates all possible values of $\mathcal{A}_i^*$ for each $i \in [k^*]$ and all possible values of $\mathcal{L}_i$ for each $i \in [k] \backslash [k^*]$. Consequently, the $k$ sets $\mathcal{A}_1^*, \cdots, \mathcal{A}_{k^*}^*, \mathcal{L}_{k^*+1}, \cdots, \mathcal{L}_k$ are guaranteed to be captured by the algorithm. Observe that Algorithm 2 associates each facility $f$ with a random integer $\eta(f) \in [k]$. It can be shown that equality $\eta(f_i^*) = i \,\forall i \in [k]$ holds with probability $k^{-k}$. When this equality is satisfied, and the $k$ sets $\mathcal{A}_1^*, \cdots, \mathcal{A}_{k^*}^*, \mathcal{L}_{k^*+1}, \cdots, \mathcal{L}_k$ are provided, the algorithm is able to construct a set $\mathcal{A}_i$ satisfying $f_i^* \in \mathcal{A}_i \subseteq \mathcal{A}_i^*$ for each $i \in [k^*]$ by extracting the facilities $f \in \mathcal{A}_i^*$ with $\eta(f) = i$ in step 14, and find a facility $f \in \mathcal{L}_i$ with $\eta(f) = i$ to construct a singleton set $\mathcal{A}_i \subseteq \mathcal{L}_i$ for each $i \in [k] \backslash [k^*]$ in step 17. Thus, the first property stated in Lemma 7 is true.

Now we consider the second property. Given a collection $\{\mathcal{A}_1, \cdots, \mathcal{A}_k\} \in \mathbb{A}$ with $\mathcal{A}_i \ne \emptyset$ for each $i \in [k]$ and a tuple $(f_1, \cdots, f_k) \in \mathcal{A}_1 \times \mathcal{A}_2 \times \cdots \times \mathcal{A}_k$, the fact that facilities in different sets are associated with distinct integers implies that $\{f_1, \cdots, f_k\}$ is a distinct set. Combining this with the decision condition employed in step 6 of Algorithm 2, we have

$$|\{i \in [k] : t \in \tau(f_i)\}| = |\{i \in [k] : t \in \mathcal{D}_i\}| \in [\alpha_t, \beta_t]$$

for each $t \in [\ell]$, where $\mathcal{D}_1, \cdots, \mathcal{D}_k$ are the $k$ sets of demographic labels used for constructing $\{\mathcal{A}_1, \cdots, \mathcal{A}_k\}$. This completes the proof of Lemma 7. $\qquad\square$

# E  Proof of Lemma 9

**Lemma 9** *The following event occurs with probability no less than $1 - e^{-1}$: Algorithm 3 yields a feasible $\mathcal{H}^\dagger$ to $\tilde{\mathcal{I}}$ satisfying $\sum_{c \in \tilde{\mathcal{C}}} w(c)\delta^\rho(c, \mathcal{H}^\dagger) < (1+4\epsilon)opt$ if $\rho = 1$ and $\sum_{c \in \tilde{\mathcal{C}}} w(c)\delta^\rho(c, \mathcal{H}^\dagger) < (1 + 9\sqrt{\epsilon})opt$ if $\rho = 2$.*

**Proof** The second property stated in Lemma 7 immediately implies that all candidate solutions constructed by Algorithm 3 are feasible solutions to $\tilde{\mathcal{I}}$. It remains to consider the approximation ratio of the algorithm. Given that Algorithm 3 iteratively invokes Algorithm 2 $k^k$ times to construct a collection $\mathbb{A}$, the probability that there exists an element $\{\mathcal{A}_1, \cdots, \mathcal{A}_k\} \in \mathbb{A}$ satisfying the first property stated in Lemma 7 is at least $1 - (1 - k^{-k})^{k^k} > 1 - e^{-1}$. For the purpose of our analysis, we assume that the collection $\mathbb{A}$ indeed contains such an element $\{\mathcal{A}_1, \cdots, \mathcal{A}_k\}$, and let $\mathcal{S}_i$ be the subset of $\mathcal{A}_i$ constructed by Algorithm 3 in step 8 or step 10 for each $i \in [k]$.

We consider an integer $i \in [k^*]$. Lemma 7 indicates that

$$\mathcal{S}_i = \mathcal{A}_i = \{f_i^*\} \tag{3}$$

if $|\mathcal{A}_i| = 1$. For the case where $|\mathcal{A}_i| > 1$, $\mathcal{S}_i$ is a $\max_{x,y \in \mathcal{A}_i} \epsilon ||x - y||$-net of $\mathcal{A}_i$. In this scenario, it can be concluded that

$$
\begin{aligned}
\delta^\rho(f_i^*, \mathcal{S}_i) &\leq \epsilon^\rho \max_{x,y \in \mathcal{A}_i} ||x - y||^\rho \\
&\leq 2\rho\epsilon^\rho \max_{f \in \mathcal{A}_i} ||f - c_i||^\rho \\
&\leq 2\rho\epsilon^\rho \cdot \max\{(1 + \epsilon)||f_i^* - c_i||^\rho, \frac{\epsilon}{n}\delta_{\max}^\rho\} \\
&\leq 2\rho\epsilon^\rho \cdot \max\{(1 + \epsilon)||f_i^* - c_i||^\rho, \frac{\epsilon}{n}opt\}, \tag{4}
\end{aligned}
$$

where the first step follows from the fact that $f_i^* \in \mathcal{A}_i$ (due to Lemma 7) and the definition of nets, the second step follows from triangle inequality (for $\rho = 1$) and Lemma 2 (for $\rho = 2$, with $\gamma = 1$), the third step follows from the fact that an integer $j \in \{0, \cdots, \lceil \epsilon^{-2} \log n \rceil\}$ satisfies $\mathcal{A}_i \subseteq \mathcal{A}^*(i, j)$ (due to Lemma 7), along with the fact that each $j \in [\lceil \epsilon^{-2} \log n \rceil]$ and $\{x, y\} \subseteq \mathcal{A}^*(i, j)$ satisfy $||x - c_i||^\rho \leq (1 + \epsilon)||y - c_i||^\rho$ and each $f \in \mathcal{A}^*(i, 0)$ satisfies $||f - c_i||^\rho \leq \epsilon\delta_{\max}^\rho n^{-1}$ (due to the definitions of $\mathcal{A}^*(i, j)$ and $\mathcal{A}^*(i, 0)$), and the last step follows from the definition of $\delta_{\max}^\rho$ and the fact that $w(c) \geq 1$ for each $c \in \tilde{\mathcal{C}}$ (due to Lemma 6).

Let $\mathbb{H}'$ be the set of candidate solution constructed using the Cartesian product $\mathcal{S}_1 \times \mathcal{S}_2 \times \cdots \times \mathcal{S}_k$ in step 11 of Algorithm 3. Equality (3) and inequality (4) suggest the existence of a candidate solution $\{f_1, \cdots, f_k\} \in \mathbb{H}'$ satisfying

$$||f_i - f_i^*||^\rho \leq 2\rho\epsilon^\rho \cdot \max\{(1 + \epsilon)||f_i^* - c_i||^\rho, \frac{\epsilon}{n}opt\} \tag{5}$$

for each $i \in [k^*]$. Thus, it can be shown that

$$
\begin{aligned}
\sum_{i=1}^{k} \sum_{c \in \tilde{\mathcal{C}}_i^*} w(c)||f_i - f_i^*||^\rho &= \sum_{i=1}^{k^*} \sum_{c \in \tilde{\mathcal{C}}_i^*} w(c)||f_i - f_i^*||^\rho \\
&\leq 2\rho\epsilon^\rho \sum_{i=1}^{k^*} \sum_{c \in \tilde{\mathcal{C}}_i^*} w(c) \max\{(1 + \epsilon)||f_i^* - c_i||^\rho, \frac{\epsilon}{n}opt\} \\
&\leq 2\rho\epsilon^\rho \sum_{i=1}^{k^*} \sum_{c \in \tilde{\mathcal{C}}_i^*} w(c) \left((1 + \epsilon)||f_i^* - c_i||^\rho + \frac{\epsilon}{n}opt\right) \\
&< 2\rho\epsilon^\rho(1 + \epsilon) \sum_{i=1}^{k} \sum_{c \in \tilde{\mathcal{C}}_i^*} w(c)||f_i^* - c_i||^\rho + 2\rho\epsilon^{\rho+1}opt \\
&\leq 2\rho\epsilon^\rho(1 + \epsilon) \sum_{i=1}^{k} \sum_{c \in \tilde{\mathcal{C}}_i^*} w(c)||f_i^* - c||^\rho + 2\rho\epsilon^{\rho+1}opt
\end{aligned}
$$

$$= 2\rho\epsilon^\rho(1+\epsilon)opt + 2\rho\epsilon^{\rho+1}opt$$
$$< 4\rho\epsilon^\rho opt, \tag{6}$$

where the first step is due to the fact that $\tilde{\mathcal{C}}_i^* = \emptyset$ for each $i \in [k]\backslash[k^*]$, the second step is due to inequality (5), the fourth step follows from the fact that $\sum_{i=1}^k \sum_{c\in\tilde{\mathcal{C}}_i^*} w(c) = \sum_{c\in\tilde{\mathcal{C}}} w(c) = |\tilde{\mathcal{C}}| < n$ (due to Lemma 6), the fifth step is due to the definition of $c_i$, the sixth step is due to the definition of $\tilde{\mathcal{C}}_i^*$, and the last step follows from the fact that $\epsilon \in (0, 0.5)$. Consequently, for the case where $\rho = 2$, we have

$$\sum_{c\in\tilde{\mathcal{C}}} w(c)\delta^\rho(c, \{f_1,\cdots,f_k\}) = \sum_{i=1}^k \sum_{c\in\tilde{\mathcal{C}}_i^*} w(c)\delta^\rho(c, \{f_1,\cdots,f_k\})$$

$$\leq \sum_{i=1}^k \sum_{c\in\tilde{\mathcal{C}}_i^*} w(c)||c - f_i||^\rho$$

$$\leq \sum_{i=1}^k \sum_{c\in\tilde{\mathcal{C}}_i^*} w(c)\left((1+\sqrt{\epsilon})||c - f_i^*||^\rho + (1+\frac{1}{\sqrt{\epsilon}})||f_i^* - f_i||^\rho\right)$$

$$= (1+\sqrt{\epsilon})opt + (1+\frac{1}{\sqrt{\epsilon}})\sum_{i=1}^k \sum_{c\in\tilde{\mathcal{C}}_i^*} w(c)||f_i^* - f_i||^\rho$$

$$< \left(1 + \sqrt{\epsilon} + 4(1+\frac{1}{\sqrt{\epsilon}})\rho\epsilon^\rho\right) opt$$

$$< (1 + 9\sqrt{\epsilon})opt, \tag{7}$$

where the third step is due to Lemma 2 (with $\gamma = \sqrt{\epsilon}$), the fourth step follows from the definition of $\tilde{\mathcal{C}}_i^*$, the fifth step follows from inequality (6), and the last step is due to the fact that $\epsilon \in (0, 0.5)$.

Replacing Lemma 2 used in the third step of inequality (7) with triangle inequality, we get

$$\sum_{c\in\tilde{\mathcal{C}}} w(c)\delta^\rho(c, \{f_1,\cdots,f_k\}) \leq \sum_{i=1}^k \sum_{c\in\tilde{\mathcal{C}}_i^*} w(c)\left(||c - f_i^*||^\rho + ||f_i^* - f_i||^\rho\right)$$

$$= opt + \sum_{i=1}^k \sum_{c\in\tilde{\mathcal{C}}_i^*} w(c)||f_i^* - f_i||^\rho$$

$$< (1 + 4\epsilon)opt \tag{8}$$

for the case where $\rho = 1$.

Using inequality (7) and inequality (8), along with the fact that Algorithm 3 returns the candidate solution with the minimum cost, we complete the proof of Lemma 9. $\qquad\square$

## F  Proof of Lemma 10

**Lemma 10** *Algorithm 3 runs in no more than $2^{(k\epsilon^{-1})^{O(1)}+k\ell}n^{O(1)}$ time.*

**Proof** Let $\mathbb{H}$ be the set of candidate solutions constructed by Algorithm 3, and let $\mathbb{A}$ be the collection formed by iteratively invoking Algorithm 2 $k^k$ times. Observe that Algorithm 2 enumerates all possible values of $k^*$, $\{c_1,\cdots,c_k\}$, $\delta_{max}^\rho$, and $\{\mathcal{L}_1,\cdots,\mathcal{L}_k\}$ to guess the sets $\{\mathcal{A}^*(1,0),\cdots,\mathcal{A}^*(k^*,\lceil\epsilon^{-2}\log n\rceil)\}$ and $\{\mathcal{L}_{k^*+1},\cdots,\mathcal{L}_k\}$. Additionally, it enumerates $[[\lceil\epsilon^{-2}\log n\rceil] \cup \{0\}]^k$ to determine the set $\{\mathcal{A}_1^*,\cdots,\mathcal{A}_{k^*}^*\}$. Analyzing the possible values of these parameters yields

$$|\mathbb{A}| \leq 2^{k\ell}k^{k+1}|\tilde{\mathcal{F}}||\tilde{\mathcal{C}}|^{k+1}(\epsilon^{-2}\log n + 1)^k$$

$$\leq 2^{k\ell}(k\epsilon^{-1}\log n)^{O(k)}n$$

$$\leq 2^{k\ell}(k\epsilon^{-1})^{O(k)}n^{O(1)}, \tag{9}$$

where the second step follows from the fact that $\tilde{\mathcal{C}} = (k\epsilon^{-1} \log n)^{O(1)}$ (due to Lemma 6) and $|\tilde{\mathcal{F}}| < n$, and the third step follows from Lemma 1 (with $s = n$ and $t = k$).

Let $\{\mathcal{A}_1, \cdots, \mathcal{A}_k\} \in \mathbb{A}$ be the collection considered in one of the iterations of steps 6-12 of Algorithm 3. Let $\mathcal{S}_1, \cdots, \mathcal{S}_k$ be the subsets constructed in step 8 or step 10 and $\mathbb{H}'$ be the set of candidate solutions constructed in step 11 during this iteration. For each $i \in [k]$ with $|\mathcal{A}_i| > 1$, $\mathcal{S}_i$ is a $\max_{x,y \in \mathcal{A}_i} \epsilon \|x - y\|$-net of $\mathcal{A}_i$, which is constructed based on the maximum distance between the facilities from $\mathcal{A}_i$ in step 10. Lemma 8 implies that constructing these nets takes no more than

$$\sum_{i=1}^{k} 2^{O(\tilde{d})} |\mathcal{A}_i|^{O(1)} \leq 2^{O(\tilde{d})} |\tilde{\mathcal{F}}|^{O(1)} k \leq 2^{O(\tilde{d})} n^{O(1)} k \tag{10}$$

time. Moreover, the upper bound on the size of a net exhibited in Lemma 8 suggests that

$$1 \leq |\mathcal{S}_i| \leq \epsilon^{-\tilde{d}} \tag{11}$$

for each $i \in [k]$. Inequality (11) leads to

$$|\mathbb{H}'| = \prod_{i=1}^{k} |\mathcal{S}_i| \leq \epsilon^{-k\tilde{d}},$$

and thus we have

$$|\mathbb{H}| \leq \epsilon^{-k\tilde{d}} |\mathbb{A}|. \tag{12}$$

Given the set $\mathbb{H}$ of candidate solutions, Algorithm 3 takes $|\tilde{\mathcal{C}}| \tilde{d} k |\mathbb{H}|$ time to identify the solution with the minimum cost. Combining this with the time required for constructing nets in each iteration exhibited in inequality (10), we know that the running time of Algorithm 3 is upper-bounded by

$$\begin{aligned}
2^{O(\tilde{d})} n^{O(1)} k |\mathbb{A}| + |\tilde{\mathcal{C}}| \tilde{d} k |\mathbb{H}| &\leq |\mathbb{A}| k (2^{O(\tilde{d})} n^{O(1)} + \epsilon^{-k\tilde{d}} |\tilde{\mathcal{C}}| \tilde{d}) \\
&\leq 2^{k\ell} \epsilon^{-O(k\tilde{d})} n^{O(1)} (k\epsilon^{-1})^{O(k)} \\
&\leq 2^{k\ell} n^{O(1)} (k\epsilon^{-1})^{O(k)} (k \log n)^{(k\epsilon^{-1})^{O(1)}} \\
&= 2^{(k\epsilon^{-1})^{O(1)} + k\ell} n^{O(1)},
\end{aligned}$$

where the first step follows from inequality (12), the second step follows from the fact that $|\tilde{\mathcal{C}}| \leq (k\epsilon^{-1} \log n)^{O(1)}$ (due to Lemma 3) and inequality (9), the third step follows from the fact that $\tilde{d} = \epsilon^{-O(1)} (\log k + \log \log n)$ (due to Lemma 3), and the last step is due to Lemma 1 (with $s = n$ and $t = (k\epsilon^{-1})^{O(1)}$). This completes the proof of Lemma 10. $\qquad \square$

## G  Proof of Theorem 1

**Theorem 1** *Given an instance $(\ell, k, \mathcal{C}, \mathcal{F}, \vec{\alpha}, \vec{\beta}, \rho, \tau)$ of fair-range clustering with $\mathcal{C} \cup \mathcal{F} \subset \mathbb{R}^d$ and $\rho \in \{1, 2\}$ along with a real number $\varepsilon \in (0, 1)$, there is a randomized $(1 + \varepsilon)$-approximation algorithm running in $O(d \log d) + 2^{(k\varepsilon^{-1})^{O(1)} + k\ell} n^{O(1)}$ time, where $n = |\mathcal{C} \cup \mathcal{F}|$.*

**Proof** Let $\mathcal{H}^{\ddagger}$ be the solution to $\mathcal{I}$ returned by Algorithm 4, and let $\mathcal{H}^{\dagger}$ be the solution to $\tilde{\mathcal{I}}$ constructed in step 2 of Algorithm 4. Since each facility in $\mathcal{H}^{\ddagger}$ shares the same set of demographic labels as its corresponding image in $\mathbb{R}^{\tilde{d}}$, it follows that $\mathcal{H}^{\ddagger}$ is a feasible solution to $\mathcal{I}$. Let $\mathcal{H}^*$ be an optimal solution to $\mathcal{I}$ and $\tilde{\mathcal{H}}^*$ be an optimal solution to $\tilde{\mathcal{I}}$.

The optimality of $\tilde{\mathcal{H}}^*$ for $\tilde{\mathcal{I}}$ and Lemma 6 imply that

$$\begin{aligned}
\sum_{c \in \tilde{\mathcal{C}}} w(c) \delta^{\rho}(c, \tilde{\mathcal{H}}^*) &\leq \sum_{c \in \tilde{\mathcal{C}}} w(c) \delta^{\rho}(c, \{\varphi(f) : f \in \mathcal{H}^*\}) \\
&\leq (1 + \epsilon)^{2\rho + 1} \sum_{c \in \mathcal{C}} \delta^{\rho}(c, \mathcal{H}^*). \tag{13}
\end{aligned}$$

For the case where $\rho = 1$, we can derive that inequality

$$\sum_{c \in \mathcal{C}} \delta^\rho(c, \mathcal{H}^\ddagger) \leq \frac{1}{1-\epsilon} \sum_{c \in \tilde{\mathcal{C}}} w(c) \delta^\rho(c, \mathcal{H}^\dagger)$$
$$< \frac{1+4\epsilon}{1-\epsilon} \sum_{c \in \tilde{\mathcal{C}}} w(c) \delta^\rho(c, \tilde{\mathcal{H}}^*)$$
$$\leq \frac{1+4\epsilon}{1-\epsilon} (1+\epsilon)^{2\rho+1} \sum_{c \in \mathcal{C}} \delta^\rho(c, \mathcal{H}^*)$$
$$< (1+39\epsilon) \sum_{c \in \mathcal{C}} \delta^\rho(c, \mathcal{H}^*) \tag{14}$$

holds with probability no less than $1 - e^{-1}$, where the first step is due to the fact that $\mathcal{H}^\dagger = \{\varphi(f) : f \in \mathcal{H}^\ddagger\}$ and Lemma 6, the second step follows from Lemma 9, the third step is due to inequality (13), and the last step follows from the fact that $\epsilon \in (0, 0.5)$. Similarly, we can conclude that inequality

$$\sum_{c \in \mathcal{C}} \delta^\rho(c, \mathcal{H}^\ddagger) \leq \frac{1}{1-\epsilon} \sum_{c \in \tilde{\mathcal{C}}} w(c) \delta^\rho(c, \mathcal{H}^\dagger)$$
$$< \frac{1+9\sqrt{\epsilon}}{1-\epsilon} \sum_{c \in \tilde{\mathcal{C}}} w(c) \delta^\rho(c, \tilde{\mathcal{H}}^*)$$
$$\leq \frac{1+9\sqrt{\epsilon}}{1-\epsilon} (1+\epsilon)^{2\rho+1} \sum_{c \in \mathcal{C}} \delta^\rho(c, \mathcal{H}^*)$$
$$< (1+83\sqrt{\epsilon}) \sum_{c \in \mathcal{C}} \delta^\rho(c, \mathcal{H}^*) \tag{15}$$

holds with the same probability when $\rho = 2$, where the second step is due to Lemma 9.

Using inequality (14) and inequality (15), we know that Algorithm 4 is a randomized $(1 + 39\epsilon)$-approximation algorithm for the FkMed problem and a $(1 + 83\sqrt{\epsilon})$-approximation algorithm for the FkMeans problem. Moreover, Lemma 6 and Lemma 10 imply that this algorithm runs in $O(d \log d) + 2^{(k\epsilon^{-1})^{O(1)} + k\ell} n^{O(1)}$ time.

Given a constant $\varepsilon \in (0, 1)$, let $\epsilon = \frac{\varepsilon}{39}$ for the FkMed problem and $\epsilon = (\frac{\varepsilon}{83})^2$ for the FkMeans problem, then the argument above implies the existence of $(1 + \varepsilon)$-approximation algorithms with running time $O(d \log d) + 2^{(k\varepsilon^{-1})^{O(1)} + k\ell} n^{O(1)}$ for both problems, as desired. $\square$

