# OpenReview forum: "Parameterized Approximation Schemes for Fair-Range Clustering"
_NeurIPS.cc/2024/Conference — NeurIPS 2024 poster_

### Official Review · Reviewer_HhQt · 2024-07-05

**Soundness:** 3
**Presentation:** 3
**Contribution:** 3
**Rating:** 8
**Confidence:** 5

**Summary:**

The authors study the fair range clustering problem, where facilities are associated with multiple demographic labels, forming intersecting groups. They impose both lower and upper bounds on the number of cluster centers chosen from each label. For both $k$-median and $k$-means clustering objectives, they present a $1 + \epsilon$ approximation algorithm when the underlying metric space is Euclidean.

The key contribution of this work is leveraging the properties of Euclidean metric spaces to improve approximation ratios while maintaining similar running times as for general metric spaces, specifically fixed-parameter tractable (FPT) in parameters $k$ and $\ell$.

The authors make use of the techniques and results that are known the literature, but it is still challenging to stitch the pieces together to obtain a solution and formally prove all the claims, which the authors have successfully managed to do. I have verified proofs in sufficient detail and cannot find anything wrong or incorrect. It is possible that I may have missed something.

The writing can be simplified by clearly explaining the figures and explicitly stating that Figure 2 applies only to clients. It took me some time to realize that the facility set does not undergo the same transformation, as the coreset applies only to clients. Also it was difficult to understand precisely what the authors refer to as the annular region in Figure 1a (this was only clear after reading Section 4.2, and discretisation of distances).

In a nutshell, the approach works by first reducing the high-dimensional space to lower-dimensions, present $1 + \epsilon)$ FPT algorithm in lower-dimensions and show that this solution translates to a $ 1 + O(\epsilon)$ FPT approximation in higher dimensions.

**Strengths:**

This work has significant theoretical contributions, offering new insights into a known difficult clustering problem variant, where earlier works have extensively investigated and established the computational complexity of the problem.

**Weaknesses:**

Similar to earlier work by Thejaswi et al. (2022), the presented FPT algorithms may not scale practically, given that the exponential factors are large. In theory, dimension reduction and coreset constructions are expected to introduce a minimal $\epsilon$ factor of distortion in distances. However, in practice, creating smaller-sized coresets often requires a larger $\epsilon$, which limits the practical scalability of algorithms. Also the achieved approximation factors in practical applications may be significantly larger than the theoretical claims. Even though the theoretical contributions are good, I would suggest that the authors put an effort to perform (at least some) experimental evaluations when submitting to an applied conference such as NeurIPS.

**Questions:**

The extension from considering only lower bound requirements to including both lower and upper bound requirements is straightforward in the work of Thejaswi et al. 2022. In their work, see Lemma~5.2, which exhaustively lists/enumerates all feasible constraint patterns satisfying the lower bound requirements, and this can be extended to enumerate feasible constrained patterns that satisfy both lower and upper bound requirements. Consequently, all algorithmic findings of diversity-aware clustering (with lower bound requirements) would extend from this adaptation to fair-range clustering (with lower and upper bound requirements), although the authors themselves do not explicitly assert this claim, the extension is immediate. Do you agree with this, or do you hold a different viewpoint?

Line 52-54: This statement is not accurate. The reduction presented by Thejaswi et al. (2021) from diversity-aware $k$-median to matroid median problem shows that any algorithm for matroid median can effectively handle the scenario with disjoint facility groups. Consequently, this reduction yields a $7.081 + \epsilon$ approximation for the disjoint facility groups case, as further detailed in Theorem~7.1 of Thejaswi et al. (2024) preprint on ArXiv this also yields a FPT($k$) algorithm with 1 + 2 e^{-1} approximation. Do you hold a different viewpoint on this?

Figure 1(a): The caption does not clearly explain the significance of the two circles around the client. It would be beneficial to explicitly specify these details. Are these circles representing the distance defined by the authors in Lines 228-229 on Page 6 (as a consequence of discretisation of distances)? please clarify.

**Limitations:**

In the broader impact section authors have briefly mentioned the potential social impact of their work.

---

> ### Author Rebuttal · Authors · 2024-08-06
>
> We thank the reviewer for the thoughtful comments. We summarize our responses in the following.
>
> **Q1: Regarding the extension from diversity-aware clustering to fair-range clustering.**
>
> Response: Thanks for pointing this out. Following your guidance, we found that when $k$ and $\ell$ (i.e., the maximal number of opened facilities and the number of demographic groups) are fixed parameters, it is straightforward to extend an algorithm for diversity-aware clustering to fair-range clustering, based on the method for enumerating feasible constraint patterns given in the work of Thejaswi et al. (2022). We will clarify this in the revised version.
>
> **Q2: Regarding the reduction to the matroid median problem.**
>
> Response: Sorry for the incomplete description of related work. Thejaswi et al. (2021) demonstrated that the diversity-aware clustering problem can be reduced to the matroid clustering problem when the demographic groups are disjoint and the sum of lower bounds associated with the groups equals $k$. It was also indicated that this reduction can be extended to the fair-range clustering problem with an $O(k^{\ell-1})$ multiplicative overhead in the running times of the algorithms. We will clarify this reduction and the corresponding approximation results in the part of related work in the revised version.
>
> **Q3: Figure 1(a): The caption does not clearly explain the significance of the two circles around the client. It would be beneficial to explicitly specify these details. Are these circles representing the distance defined by the authors in Lines 228-229 on Page 6 (as a consequence of discretisation of distances)? Please clarify.**
>
> Response: In Lines 228-229, we introduce a set of annuli centered at $c_i$ by discretizing the distances from the facilities to $c_i$. Each annulus is defined such that its outer radius is $1+\varepsilon$ times its inner radius. The circles shown in Figure 1(a) represent the outer and inner circles of the annulus involving $f^*_i$ (i.e., the facility corresponding to the leader $c_i$). This will be clarified in the revised version.
>
> **Q4: Regarding the explanation of Figure 2.**
>
> Response: Sorry for the unclear presentation. In the revised version, we will include a description of the process illustrated in Figure 2. Additionally, we will enhance the clarity of the data-reduction algorithm by explaining the roles of JL-transform and coreset construction.

---

> ### Comment · Reviewer_HhQt · 2024-08-08
> **Response to rebuttal**
>
> Thank for your responses. I agree with the authors' replies and will update my evaluation accordingly.
>
> However, the authors have not addressed the identified weaknesses. If they acknowledge these issues, they should discuss them in the revised version of the paper. If they disagree, I recommend presenting counterarguments to address these concerns.

---

> > ### Author Response · Authors · 2024-08-08
> > **Thank you**
> >
> > Many thanks for reading our response and appreciation of our work.
> >
> > In the revised version, we will carefully discuss the practical impact of the work according to your comments. On one hand, we will further illustrate the widespread consideration of Euclidean data in practical clustering tasks, underscoring our work's contribution to revealing different approximability of the Euclidean version of the problem, especially in the context of the larger tight ratios in general metric spaces. On the other hand, when a constant loss in the approximation ratio is acceptable, small coresets for $k$-median and $k$-means clustering can be constructed using fast bi-criteria approximation algorithms (such as the ones based on $k$-means++ [1]). By allowing a sacrifice in the approximation ratio, we believe that our coreset-based adaptation of the JL-transform has the potential to accelerate existing heuristics in high-dimensional Euclidean spaces, including those proposed by Thejaswi et al. (2022).
> >
> > [1] Ankit Aggarwal, Amit Deshpande, and Ravi Kannan. Adaptive Sampling for $k$-Means Clustering. In Proc. of APPROX-RANDOM 2009: 15-28

---

### Official Review · Reviewer_py4M · 2024-07-11

**Soundness:** 3
**Presentation:** 3
**Contribution:** 3
**Rating:** 6
**Confidence:** 3

**Summary:**

The paper presents fixed-parameter approximation schemes for the fair-range k-median and k-means problems in Euclidean spaces, parameterized by both the number of facilities and labels. The results improve on existing results, which could only achieve constant-ratio approximation. The main technique used is a data-reduction technique to reduce the dimensionality, combined with an algorithm for low-dimensional spaces.

**Strengths:**

1. The paper presents an FPT APX-scheme for very important clustering problems.
2. The approximation results improve on the current best results.
3. The results are theoretically solid and technical.

**Weaknesses:**

1. The techniques are not novel.
2. The results are purely theoretical, with no experimental results provided.

**Questions:**

Did you consider any other notions of fairness, based on other constraints (e.g., related to clients)?

**Limitations:**

Lack of experimental work.

---

> ### Author Rebuttal · Authors · 2024-08-06
>
> We thank the reviewer for the thoughtful comments. We summarize our responses in the following.
>
> **Q1: Did you consider any other notions of fairness, based on other constraints (e.g., related to clients)?**
>
> Response: Thanks for the question. For the fair clustering problem where clients are partitioned into different demographic groups and the proportion of each group in each cluster is constrained, it is known that a variant of the algorithm stated in Lemma 3 yields small-size coresets [1]. By replacing the algorithm in Lemma 3 with this variant, one can generalize our data-reduction method (Algorithm 1) to address the clustering problem with the fairness constraint imposed on clients. Consequently, we believe that the ideas in the paper have the potential of being applicable to the client-constrained case. We will add this intuition and related work on the client-constrained fair clustering problem in the revised version.
>
> [1] Sayan Bandyapadhyay, Fedor V. Fomin, and Kirill Simonov. On Coresets for Fair Clustering in Metric and Euclidean Spaces and Their Applications. In Proc. of ICALP 2021: 23:1-23:15
>
> **Q2: Regarding the novelty of the work.**
>
> Response: Some aspects of the work are built upon existing methods, such as Johnson-Lindenstrauss transform and the construction of coresets and nets. However, leveraging these well-known techniques to improve upon the previously known constant-factor approximation ratios in Euclidean spaces is non-trivial. To achieve this goal, we give a novel approach for exploring the properties of the Euclidean metric in the context of the fair-range clustering problem. This includes partitioning the solution search space into small cells and carefully balancing the number of cells (which affects the running time) with the distance from each facility opened in an optimum to the center point of the cell it belongs to (which affects the approximation ratio). In our revised version, we will make the ideas that distinguish the work clearer in the introduction.
>
> **Q3: Regarding the nonexistence of experimental work.**
>
> Response: Our research focuses on the theoretical aspect of the fair-range clustering problem, demonstrating an upper bound of $1+\varepsilon$ on the parameterized approximation ratio in Euclidean spaces. Considering the significant attention received by the hardness and approximability of the problem [2,3,4], we believe that gaining this new understanding of its approximability is of independent interest.
>
> [2] Suhas Thejaswi, Ameet Gadekar, Bruno Ordozgoiti, and Aristides Gionis.
> Diversity-Aware Clustering: Computational Complexity and Approximation Algorithms. CoRR abs/2401.05502, 2024
>
> [3] Zhen Zhang, Junfeng Yang, Limei Liu, Xuesong Xu, Guozhen Rong, and Qilong Feng. Towards a Theoretical Understanding of Why Local Search Works for Clustering with Fair-Center Representation. In Proc. of AAAI 2024: 16953-16960
>
> [4] Sèdjro Salomon Hotegni, Sepideh Mahabadi, and Ali Vakilian.
> Approximation Algorithms for Fair Range Clustering. In Proc. of ICML 2023: 13270-13284

---

> > ### Comment · Reviewer_py4M · 2024-08-10
> >
> > Thank you for your response. I will keep my score.

---

> > > ### Author Response · Authors · 2024-08-10
> > > **Thank you**
> > >
> > > Many thanks for reading our response and continued support.

---

### Official Review · Reviewer_c25h · 2024-07-12

**Soundness:** 3
**Presentation:** 2
**Contribution:** 3
**Rating:** 5
**Confidence:** 3

**Summary:**

- This paper studies fair range clustering, which aims to ensure that cluster centers are not dominated by specific demographic groups.
- It focuses on fair range clustering using the 1-norm and 2-norm distance metrics.
- The paper proposes an algorithm with three internal steps: (i) data reduction to low-dimensional space, (ii) obtaining fair centers in the low-dimensional space, and (iii) transforming the centers back to the original high-dimensional space.
- The authors theoretically demonstrate that the algorithm can be computed in Fixed Parameter Tractable (FPT) time.

**Strengths:**

- The problem considered in this paper allows each center (facility) to include multiple demographics, whereas a previous work considers a single demographic label for each center.
- The proposed algorithm can be computed in FPT-time.
- The algorithm is theoretically applicable to high-dimensional data.

**Weaknesses:**

1. The writing and presentation are complex, making it difficult for non-experts to follow. It assumes readers are very familiar with fair range clustering.
2. The paper focuses too much on techniques used to derive the algorithms, lacking intuitive explanations about how the algorithm works.
3. The introduction should better justify the necessity of fair range clustering. It should include related works on (conventional) fair clustering [A, B, C] and their key differences to highlight the significance of fair "range" clustering.
4. The data reduction mechanism lacks novelty, appearing to be a direct consequence of combining Lemmas 3, 4, and 5, which were not developed by the authors.
5. The link mapping $\phi$ between low-dimensional space and high-dimensional space, which is a key component of the proposed algorithm, is theoretically valid (through Lemmas 4 and 5), but practical construction methods are not discussed.
6. Numerical analysis is missing. Given existing works on fair clustering [A, B, C] or fair range clustering [D] with numerical results on real data, appropriate experiments are necessary.
7. (Minor) line 451: second -> third, line 454 : third -> fourth

[A] Fair Clustering Through Fairlets
https://dl.acm.org/doi/pdf/10.5555/3295222.3295256

[B] Fair Algorithms for Clustering
https://proceedings.neurips.cc/paper_files/paper/2019/file/fc192b0c0d270dbf41870a63a8c76c2f-Paper.pdf

[C] Variational Fair Clustering
https://arxiv.org/abs/1906.08207

[D] Fair k-Center Clustering for Data Summarization
https://proceedings.mlr.press/v97/kleindessner19a/kleindessner19a.pdf

**Questions:**

1. What is the intuitive definition of “core set”?
2. What is the role of the core set in data reduction?
3. The authors mention that Thejaswi et al. [2022] showed limits of approximation order in a general metric space (lines 73-77). How is the approximation order improved when considering Euclidean space rather than a general metric space?
4. Is $\tilde{\mathcal{S}^*}$ in line 504 defined? If not, what is its definition?

**Limitations:**

1. It is not confirmed that the proposed algorithm can be applied to real scenarios.

---

> ### Author Rebuttal · Authors · 2024-08-06
>
> We thank the reviewer for the thoughtful comments. We summarize our responses in the following.
>
> **Q1: What is the intuitive definition of ''core set''?**
>
> Response: Thanks for the question. A ''core set'' is a small subset of the client set. For any feasible solution, its costs on the original instance and reduced instance (where the client set is replaced by the core set) are approximately the same. Constructing a core set and using it to reduce the instance can significantly enhance computational efficiency while ensuring that the solutions closely match those obtained from the full client set. We will clarify this in our revised version.
>
> **Q2: What is the role of the core set in data reduction?**
>
> Response: In data reduction, constructing a core set helps to reduce the size of the client set: We replace the client set with a smaller core set. This has the following two effects.
>
> (1) When invoking Lemma 5 with the client set as input, we can map the instance into a low-dimensional space without significantly altering the distance between each pair of client and facility. Here, the dimension is logarithmically related to the number of clients. By reducing the number of clients using a core set, we can significantly lower the dimension.
>
> (2) We construct the solution based on the ''leaders'' (i.e., the clients closest to the facilities opened in an optimal solution) within the client set. Replacing the original client set with the core set allows us to identify these leaders in FPT time.
>
> We will clarify the role of the core set in our revised version.
>
> **Q3: The authors mention that Thejaswi et al. [2022] showed limits of approximation order in a general metric space (lines 73-77). How is the approximation order improved when considering Euclidean space rather than a general metric space?**
>
> Response: The properties of the Euclidean metric allow us to carefully partition the space. We can thus restrict the solution search space to a more refined range, and thereby break through the lower bound on the approximation ratio that applies to general metric spaces. Specifically, our algorithm partitions the search space into smaller cells, ensuring that each facility opened in an optimum is close to the center point of the cell it belongs to. Consequently, we can identify a nearly-optimal solution by enumerating the subsets of center points of the cells. We will describe this idea in Section 1.1 of the revised version.
>
> **Q4: Is $\tilde{\mathcal{S}}^{*}$ in line 504 defined? If not, what is its definition?**
>
> Response: The notation $\tilde{\mathcal{S}}^*$ should be $\tilde{\mathcal{H}}^*$, which denotes an optimal solution to $\tilde{\mathcal{I}}$. Sorry for the typo.
>
> **Q5: Regarding the presentation.**
>
> Response: Thanks for pointing out the issues with the presentation. We apologize for being unable to response to each issue individually due to the space constraint. We will carefully correct the typos and reorganize the content in the revised version. First, we will clarify the intuitive ideas of our algorithms, including the roles of Lemmata 3, 4, and 5, as well as the partitioning of the solution search space. Second, we will add related work on conventional fair clustering, and underline the significance of fair range clustering by highlighting its important roles in applications where the centers are required to fairly represent the demographic groups.
>
> **Q6: The data reduction mechanism lacks novelty, appearing to be a direct consequence of combining Lemmas 3, 4, and 5, which were not developed by the authors.**
>
> Response: JL-transform (Lemmas 4 and 5) has been widely utilized in clustering problems. However, given that most previous works directly use this transform to construct an $O(\log n)$-dimensional space, we think it is somewhat unexpected that this transform, combined with a coreset-construction method (Lemma 3), yields an $O(\log k+\log\log n)$-dimensional space. This allows us to partition the solution search space into sufficiently small cells. Furthermore, given that there are algorithms for constructing coresets for the fair clustering problem where the fairness constraint is imposed on the clients [1], our idea for data reduction has the potential of being applicable to this problem as well.
>
> [1] Sayan Bandyapadhyay, Fedor V. Fomin, and Kirill Simonov. On Coresets for Fair Clustering in Metric and Euclidean Spaces and Their Applications. In Proc. of ICALP 2021: 23:1-23:15
>
> **Q7: Regarding the practicality of the mapping $\phi$.**
>
> Response: The coreset-construction algorithm in Lemma 3 runs in linear time, and one can easily trade off the size of the constructed coreset against its approximation guarantee. Moreover, it is built upon random sampling and is simple to implement. Thus, we believe that our coreset-based adaption of JL-transform is quite practical.
>
> **Q8: Regarding the nonexistence of experiments.**
>
> Response: In line with recent advancements in hardness and approximability of fair range clustering [2,3,4], our research focuses on theoretical aspects, breaking through the lower bound in general metric spaces and providing the first FPT approximation scheme under the Euclidean metric. Given the widespread consideration of Euclidean data in fair range clustering problems, we believe that gaining such a new understanding of the problem's approximability in Euclidean spaces is of intrinsic value.
>
> [2] Suhas Thejaswi, Ameet Gadekar, Bruno Ordozgoiti, and Aristides Gionis. Diversity-Aware Clustering: Computational Complexity and Approximation Algorithms. CoRR abs/2401.05502, 2024
>
> [3] Zhen Zhang, Junfeng Yang, Limei Liu, Xuesong Xu, Guozhen Rong, and Qilong Feng. Towards a Theoretical Understanding of Why Local Search Works for Clustering with Fair-Center Representation. In Proc. of AAAI 2024: 16953-16960
>
> [4] Sèdjro Salomon Hotegni, Sepideh Mahabadi, and Ali Vakilian.
> Approximation Algorithms for Fair Range Clustering. In Proc. of ICML 2023: 13270-13284

---

> > ### Comment · Reviewer_c25h · 2024-08-12
> > **Thank you for the responses**
> >
> > I appreciate the authors' rebuttal responses.
> > In light of these responses, I will raise the score.
> > However, I suggest further improvements to the presentation, especially for non-experts.
> > Specifically, it would be beneficial to (1) clarify the definition and necessity of fair range clustering in the earlier parts of the paper, (2) provide intuitive explanations of key techniques (e.g., the definition of the coreset, how to find $\phi$), and (3) offer a detailed exposition of Algorithm 2, as mentioned by other reviewers.

---

> > > ### Author Response · Authors · 2024-08-12
> > > **Thank you**
> > >
> > > Many thanks for raising the score and your further insightful guidance.
> > >
> > > We will carefully improve the presentation in the revised version, making our work more understandable according to your guidance.

---

### Official Review · Reviewer_nDwN · 2024-07-12

**Soundness:** 3
**Presentation:** 4
**Contribution:** 3
**Rating:** 7
**Confidence:** 4

**Summary:**

The paper deals with the problem of fair-range clustering in Euclidean metric spaces. In fair-range clustering, one is given a set of clients and a set of possible facilities. Every facility is associated with a subset of $\ell$ many classes. The goal is to pick up to $k$ many facilities such that a given clustering objective is minimized with respect to the chosen facilities as centers under the side constraint that the number of facilities that are associated with a certain class lies $i$ within a given interval $[\alpha_i,\beta_i]$, for all $i\leq \ell$.
The clustering objectives considered here are the $k$-Median and the $k$-Means objective.
The authors propose FPT $(1+\epsilon)$-approximations for both problems with parameters $k$ and $\ell$.
Their approach essentially works by first mapping the input to an Euclidean space of lower dimension and computing a coreset. For such a low-dimensional weighted instance, they give an algorithm that computes a near-optimal approximation on the modified instance. In the end, they translate the solution back to a solution for the original instance by returning the pre-image of the injective mapping from the transformation step.

**Strengths:**

This is a well-written and clear paper with a good contribution. There already existed optimal FPT($k$,$\ell$)-algorithms for the two considered problems matching known lower bounds. In this paper, the authors manage to break the barriers for general metric spaces when restricting to Euclidean metric spaces by giving near-optimal $(1+O(\epsilon))$-approximation algorithms for the Euclidean setting.

**Weaknesses:**

Apart from the Questions mentioned later, I only have minor comments. I am willing to increase my score to "accept" when addressing the questions.

_Comments to the authors:_
- please provide a citation for the results mentioned in the abstract (lines 9-10)
- please define $\tau$ and $\ell$ in the preliminaries
- are Lemma 1 and Lemma 2 needed somewhere in the main body of the paper? Otherwise, they could be transferred to the appendix
- figure 1: please provide an explanation of the different symbols (small circles = clients, squares = facilities?)
- l. 171: what do you mean by "distance parameter"?
- why is the upper bound stated in equation (2) meaningful? How does it help in bounding $\delta_{\max}^{\rho}$ if it contains $\delta_{\max}^{\rho}$?
- how are the annuli guessed if the possible radii depend on the maximal distance within an optimal solution in lines 229,230? When looking at the proof of Lemma 8, it becomes obvious that the value of $\delta_{\max}^{\rho}$ is guessed. Please also provide this information in the text.
- In general, is a bit difficult to recognize which are the parts that are guessed in Algorithm 2. If I understood correctly, the rings $A_i$ are guessed (which includes guessing the leader $c_i$, the radius of the annulus, which is based on guessing $\delta_{\max}^{\rho}$, and the set $L_i$), the color coding is done randomly, and the feasible solution of lowest cost is again guessed (lines 15-16 in Algorithm 2). It would be helpful to specifically state which things have to be guessed in the text.
- It is not specified how the values are chosen randomly in line 9 of Algorithm 2, resp. lines 249-253 (Uniformly at random?). Please make this more formal.
- it does not become entirely clear to me how the feasible solution is chosen in lines 15-16 in Algorithm 2. Are you iterating over *all* possible subsets of $S$ and only then check whether the subset contains at most $k$ elements? This seems a bit wasteful. Why would it not be enough to choose one facility from every ring? It would be helpful if you write this part in lines 258-259 a bit clearer.



_Language, Typos & Layout_
- 112: "number" -> "numbers"
- 216: I guess that "see Appendix [...]" belongs in the next line
- 223: "denotes" -> "denote"
- 229, 230: Why do you use $\|f-c_i\|^{\rho}$ instead of $\delta^{\rho}(f,c_i)$ here?
- it might increase readability if you restate Theorem 1 where it is proven

**Questions:**

- line 9 of Algorithm 2 assigns every facility an integer at random. This makes your algorithm a randomized algorithm and influences the result of Lemma 9. As Lemma 9 is used in the proof of Theorem 1, Theorem 1 should also mention that this guarantee holds *with high probability*. Please clarify.
- What is the motivation for using first the original Lindenstrauss transform and then the stronger version in the second step? Why do you not need the stronger transform in the first step?

**Limitations:**

Yes, the authors clearly state the limitations (fixed-parameter tractability with parameters $k$ and $\ell$, restriction to Euclidean metric spaces) in the abstract. In Section 6, they also state considerations to make when using their results for practical applications.

---

> ### Author Rebuttal · Authors · 2024-08-06
>
> We thank the reviewer for the thoughtful comments. We summarize our responses in the following.
>
> **Q1: Regarding the randomness of Algorithm 2.**
>
> Response: Thanks for pointing this out. The probability that Algorithm 2 yields the desired $(1+\varepsilon)$-approximation solution is the same as the probability that inequalities in Lemma 9 hold. In our revised version, we will modify Theorem 1 and its proof to clarify the randomness of our algorithm and the associated success probability.
>
> **Q2: What is the motivation for using first the original Lindenstrauss transform and then the stronger version in the second step? Why do you not need the stronger transform in the first step?**
>
> Response: The motivation for using the original transform in the first step is primarily driven by computational efficiency: Although the stronger transform guarantees that the distance similarity between the original space and its low-dimensional mapping is preserved over a broader range, it involves semidefinite programming and has a higher time complexity.
>
> In the first step, we use the union of clients and facilities as the input for the dimension-reduction method, aiming to map the instance to a space whose dimension is independent of $d$ (the dimension of the original space). This mapping needs to preserve the distance between each pair of points in the input set, without considering the points outside this set. Both the original and stronger transforms can achieve this goal. However, we choose the original transform because it has a lower time complexity.
>
> In the second step, we use a coreset of size $(k\log n)^{O(1)}$ as the input for the dimension-reduction method, aiming to map the instance to the desired $O(\log k+\log\log n)$-dimensional space. Here, we need to preserve the distance between any point in the coreset and any facility (outside the coreset). Due to the requirement of preserving distances over a broader range than in the first step, we choose the stronger transform in this step.
>
> We will elaborate on the above intuition in our revised version to enhance the understanding of Algorithm 1.
>
> **Q3: Regarding the presentation.**
>
> Response: We thank the reviewer for pointing out the typos and issues with the presentation. We will carefully correct these in our revised version. We apologize for being unable to response to each typo and presentation issue individually due to the space constraint.
>
> **Q4: Line 171: what do you mean by ''distance parameter''?**
>
> Response: The distance parameter denotes the parameter ''$\lambda$'' in Definition 2 and Lemma 6. As this parameter decreases, the constructed net becomes larger, causing our algorithm to take more time, but the opened facilities selected from the net more closely approximate the optimal ones. We will make this clear in the revised version.
>
> **Q5: Why is the upper bound stated in equation (2) meaningful? How does it help in bounding $\delta^\rho_{\max}$ if it contains $\delta^\rho_{\max}$?**
>
> Response: We construct annuli centered at the leaders. Since the left-hand side of equation (2) is the maximum radius of the annuli (recall that the distance from each facility in the annulus $A(i,j)$ to the leader $c_i$ is at most $\epsilon(1+\epsilon)^{j}\delta^\rho_{\max}n^{-1}$ and $j\in\\{0,\cdots,\lceil\epsilon^{-2}\log n\rceil\\}$), and the right-hand side denotes the maximum distance among the facilities in $\tilde{H}^*$ to their corresponding leaders, equation (2) implies that each facility in $\tilde{H}^*$ is involved in one of the annuli. This suggests that enumerating over $j\in\\{0,\cdots,\lceil\epsilon^{-2}\log n\rceil\\}$ to identify an annulus $A(i,j)$ that contains the optimal opened facility $f_i^*$ is feasible. We will clarify this in the revised version.
>
> **Q6: How are the annuli guessed if the possible radii depend on the maximal distance within an optimal solution in lines 229, 230? When looking at the proof of Lemma 8, it becomes obvious that the value of $\delta^\rho_{\max}$ is guessed. Please also provide this information in the text.**
>
> Response: We guess the value of $\delta^\rho_{\max}$ by examining the distance between each pair of client and facility. We need to enumerate $n^{O(1)}$ items in this process, which introduces a $n^{O(1)}$ multiplicative overhead in the running time of the algorithm. We will make this clear in the revised version.
>
> **Q7: Regarding the parts that are guessed in Algorithm 2.**
>
> Response: Algorithm 2 guesses the rings $A_i$ by enumerating the possible values of $c_i$, $L_i$, $\delta^\rho_{\max}$, and the integer $j\in\\{0,\cdots,\lceil\epsilon^{-2}\log n\rceil\\}$ satisfying $f^*_i\in A(i,j)$, iteratively performs color coding and guesses a successful iteration (which exists with constant probability), and selects the solution with lowest cost from the candidate set. We will clarify this in the description of Algorithm 2 in the revised version. Moreover, we will include an algorithm that formally describes how to construct the set $\mathbb{A}$ of possible values for $\\{A_1, \cdots, A_k\\}$, rather than providing it implicitly in the proof of Lemma 8.
>
> **Q8: It is not specified how the values are chosen randomly in line 9 of Algorithm 2, resp. lines 249-253 (Uniformly at random?). Please make this more formal.**
>
> Response: Sorry for the informal presentation. The color of each facility is chosen uniformly at random in step 7. We will make corresponding modifications in the revised version.
>
> **Q9: Regarding the selection of the feasible solution in lines 15-16 of Algorithm 2.**
>
> Response: Indeed, merging the sets of candidate opened facilities selected from the $k$ rings and enumerating the subsets of the union $S$ to find feasible solutions is unnecessary. It is sufficient to separately construct $k$ sets of candidates based on the rings and select one opened facility from each set. We will make corresponding modifications in Algorithm 2 and its description.

---

> > ### Comment · Reviewer_nDwN · 2024-08-09
> >
> > Thank you for the detailed response. All my questions are resolved and I will increase my score accordingly.

---

> > > ### Author Response · Authors · 2024-08-09
> > > **Thank you**
> > >
> > > Many thanks for reading our response and your positive rating.

---

### Author Rebuttal · Authors · 2024-08-06

We greatly appreciate the reviewers for the in-depth reviews, which have significantly helped us in improving our work. Below, we provide detailed responses to the comments.

---

### Decision · Program_Chairs · 2024-09-25

**Decision:**

Accept (poster)

**Comment:**

Fair-range clustering is a natural clustering problem that has gained attention in recent years. All reviewers agree that this submission makes significant theoretical progress on the approximability of this problem by deriving an FPT approximation scheme for the Euclidean case despite existing hardness results for general metrics. There has been a detailed discussion and the authors addressed all questions raised by the reviewers. The only weakness is the lack of an experimental evaluation.